# On Path Integration of Grid Cells:
# Group Representation and Isotropic Scaling

**Ruiqi Gao**[1*]
ruiqigao@ucla.edu

**Jianwen Xie**[2]
jianwen@ucla.edu

**Xue-Xin Wei**[3]
weixx@utexas.edu

**Song-Chun Zhu**[1,4,5]
sczhu@stat.ucla.edu

**Ying Nian Wu**[1]
ywu@stat.ucla.edu

[1]Department of Statistics, UCLA    [2]Cognitive Computing Lab, Baidu Research
[3]Department of Neuroscience, UT Austin    [4]Department of Computer Science, UCLA
[5]Beijing Institute for General Artificial Intelligence (BIGAI)

## Abstract

Understanding how grid cells perform path integration calculations remains a fundamental problem. In this paper, we conduct theoretical analysis of a general representation model of path integration by grid cells, where the 2D self-position is encoded as a higher dimensional vector, and the 2D self-motion is represented by a general transformation of the vector. We identify two conditions on the transformation. One is a group representation condition that is necessary for path integration. The other is an isotropic scaling condition that ensures locally conformal embedding, so that the error in the vector representation translates conformally to the error in the 2D self-position. Then we investigate the simplest transformation, i.e., the linear transformation, uncover its explicit algebraic and geometric structure as matrix Lie group of rotation, and explore the connection between the isotropic scaling condition and a special class of hexagon grid patterns. Finally, with our optimization-based approach, we manage to learn hexagon grid patterns that share similar properties of the grid cells in the rodent brain. The learned model is capable of accurate long distance path integration. Code is available at https://github.com/ruiqigao/grid-cell-path.

## 1 Introduction

Imagine walking in the darkness. Purely based on the sense of self-motion, one can gain a sense of self-position by integrating the self motion - a process often referred to as path integration [10, 14, 21, 15, 27]. While the exact neural underpinning of path integration remains unclear, it has been hypothesized that the grid cells [21, 17, 40, 24, 23, 12] in the mammalian medial entorhinal cortex (mEC) may be involved in this process [20, 30, 22]. The grid cells are so named because individual neurons exhibit striking firing patterns that form hexagonal grids when the agent (such as a rat) navigates in a 2D open field [18, 21, 16, 6, 34, 5, 7, 11, 29, 1]. The grid cells also interact with the place cells in the hippocampus [28]. Unlike a grid cell that fires at the vertices of a lattice, a place cell often fires at a single (or a few) locations.

The purpose of this paper is to understand how the grid cells may perform path integration calculations. We study a general optimization-based representational model in which the 2D self-position is

---

[*]The author is now a Research Scientist at Google Brain team.

represented by a higher dimensional vector and the 2D self-motion is represented by a transformation of the vector. The vector representation can be considered position encoding or position embedding, where the elements of the vector may be interpreted as activities of a population of grid cells. The transformation can be realized by a recurrent network that acts on the vector. Our focus is to study the properties of the transformation.

Specifically, we identify two conditions for the transformation: a group representation condition and an isotropic scaling condition, under which we demonstrate that the local neighborhood around each self-position in the 2D physical space is embedded conformally as a 2D neighborhood around the vector representation of the self-position in the neural space.

We then investigate the simplest special case of the transformation, i.e., linear transformation, that forms a matrix Lie group of rotation, under which case we show that the isotropic scaling condition is connected to a special class of hexagonal grid patterns. Our numerical experiments demonstrate that our model learns clear hexagon grid patterns of multiple scales which share observed properties of the grid cells in the rodent brain, by optimizing a simple loss function. The learned model is also capable of accurate long distance path integration.

**Contributions**. Our work contributes to understanding the grid cells from the perspective of representation learning. We conduct theoretical analysis of (1) general transformation for path integration by identifying two key conditions and a local conformal embedding property, (2) linear transformation by revealing the algebraic and geometric structure and connecting the isotropic scaling condition and a special class of hexagon grid patterns, and (3) integration of linear transformation model and linear basis expansion model via unitary group representation theory. Experimentally we learn clear hexagon grid patterns that are consistent with biological observations, and the learned model is capable of accurate path integration.

## 2 General transformation

### 2.1 Position embedding

Consider an agent (e.g., a rat) navigating within a 2D open field. Let $x = (x_1, x_2)$ be the self-position of the agent. We assume that the self-position $x$ in the 2D physical space is represented by the response activities of a population of $d$ neurons (e.g., $d = 200$), which form a vector $v(x) = (v_i(x), i = 1, ..., d)^\top$ in the $d$-dimensional "neural space", with each element $v_i(x)$ representing the firing rate of one neuron when the animal is at location $x$.

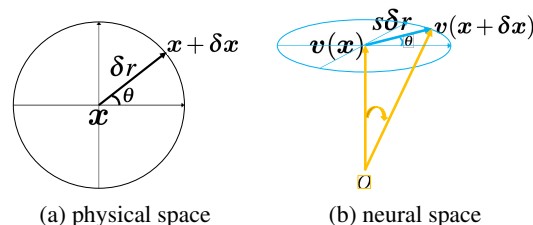

(a) physical space      (b) neural space

Figure 1: The local 2D polar system around self-position $x$ in the 2D physical space (a) is embedded conformally as a 2D polar system around vector $v(x)$ in the $d$-dimensional neural space (b), with a scaling factor $s$ (so that $\delta r$ in the physical space becomes $s\delta r$ in the neural space while the angle $\theta$ is preserved).

$v(x)$ can be called position encoding or position embedding. Collectively, $(v(x), \forall x)$ forms a *codebook* of $x \in \mathbb{R}^2$, and $(v(x), \forall x)$ is a *2D manifold* in the $d$-dimensional neural space, i.e., globally we embed $\mathbb{R}^2$ as a 2D manifold in the neural space. Locally, we identify two conditions under which the 2D local neighborhood around each $x$ is embedded *conformally* as a 2D neighborhood around $v(x)$ with a scaling factor. See Fig. 1. As shown in Section 3.3, the conformal embedding is connected to the hexagon grid patterns.

### 2.2 Transformation and path integration

At self-position $x$, if the agent makes a self-motion $\Delta x = (\Delta x_1, \Delta x_2)$, then it moves to $x + \Delta x$. Correspondingly, the vector representation $v(x)$ is transformed to $v(x + \Delta x)$. The general form of the transformation can be formulated as:

$$v(x + \Delta x) = F(v(x), \Delta x). \tag{1}$$

The transformation $F(\cdot, \Delta x)$ can be considered a representation of $\Delta x$, which forms a 2D additive group. We call Eq. (1) the *transformation model*. It can be implemented by a recurrent network to

derive a path integration model: if the agent starts from $x_0$, and makes a sequence of moves $(\Delta x_t, t = 1, ..., T)$, then the vector is updated by $v_t = F(v_{t-1}, \Delta x_t)$, where $v_0 = v(x_0)$, and $t = 1, ..., T$.

## 2.3 Group representation condition

The solution to the transformation model (Eq. (1)) should satisfy the following condition.

**Condition 1.** *(Group representation condition)* $(v(x), \forall x)$ *and* $(F(\cdot, \Delta x), \forall \Delta x)$ *form a representation of the 2D additive Euclidean group* $\mathbb{R}^2$ *in the sense that*

$$F(v(x), 0) = v(x), \ \forall x; \tag{2}$$
$$F(v(x), \Delta x_1 + \Delta x_2) = F(F(v(x), \Delta x_1), \Delta x_2), \ \forall x, \Delta x_1, \Delta x_2. \tag{3}$$

$(F(\cdot, \Delta x), \forall \Delta x)$ is a Lie group of transformations acting on the codebook manifold $(v(x), \forall x)$.

The reason for (2) is that if $\Delta x = 0$, then $F(\cdot, 0)$ should be the identity transformation. Thus the codebook manifold $(v(x), \forall x)$ consists of fixed points of the transformation $F(\cdot, 0)$. If $F(\cdot, 0)$ is furthermore a contraction around $(v(x), \forall x)$, then $(v(x), \forall x)$ are the attractor points.

The reason for (3) is that the agent can move in one step by $\Delta x_1 + \Delta x_2$, or first move by $\Delta x_1$, and then move by $\Delta x_2$. Both paths would end up at the same $x + \Delta x_1 + \Delta x_2$, which is represented by the same $v(x + \Delta x_1 + \Delta x_2)$.

The group representation condition is a necessary self-consistent condition for the transformation model (Eq. (1)).

## 2.4 Egocentric self-motion

Self-motion $\Delta x$ can also be parametrized egocentrically as $(\Delta r, \theta)$, where $\Delta r$ is the displacement along the direction $\theta \in [0, 2\pi]$, so that $\Delta x = (\Delta x_1 = \Delta r \cos\theta, \Delta x_2 = \Delta r \sin\theta)$. The egocentric self-motion may be more biologically plausible where $\theta$ is encoded by head direction, and $\Delta r$ can be interpreted as the speed along direction $\theta$. The transformation model then becomes

$$v(x + \Delta x) = F(v(x), \Delta r, \theta), \tag{4}$$

where we continue to use $F(\cdot)$ for the transformation (with slight abuse of notation). $(\Delta r, \theta)$ form a polar coordinate system around $x$.

## 2.5 Infinitesimal self-motion and directional derivative

In this subsection, we derive the transformation model for infinitesimal self-motion. While we use $\Delta x$ or $\Delta r$ to denote finite (non-infinitesimal) self-motion, we use $\delta x$ or $\delta r$ to denote infinitesimal self-motion. At self-position $x$, for an infinitesimal displacement $\delta r$ along direction $\theta$, $\delta x = (\delta x_1 = \delta r \cos\theta, \delta x_2 = \delta r \sin\theta)$. See Fig. 1 (a) for an illustration. Given that $\delta r$ is infinitesimal, for any fixed $\theta$, a first order Taylor expansion of $F(v(x), \delta r, \theta)$ with respect to $\delta r$ gives us

$$v(x + \delta x) = F(v(x), \delta r, \theta) = F(v(x), 0, \theta) + F'(v(x), 0, \theta)\delta r + o(\delta r)$$
$$= v(x) + f_\theta(v(x))\delta r + o(\delta r), \tag{5}$$

where $F(v(x), 0, \theta) = v(x)$ according to Condition 1, and $f_\theta(v(x)) := F'(v(x), 0, \theta)$ is the first derivative of $F(v(x), \Delta r, \theta)$ with respect to $\Delta r$ at $\Delta r = 0$. $f_\theta(v(x))$ is the *directional derivative* of $F(\cdot)$ at self-position $x$ and direction $\theta$.

For a fixed $\theta$, $(F(\cdot, \Delta r, \theta), \forall \Delta r)$ forms a one-parameter Lie group of transformations, and $f_\theta(\cdot)$ is the generator of its Lie algebra.

## 2.6 Isotropic scaling condition

With the directional derivative, we define the second condition as follows, which leads to locally conformal embedding and is connected to hexagon grid pattern.

**Condition 2.** *(Isotropic scaling condition) For any fixed* $x$, $\|f_\theta(v(x))\|$ *is constant over* $\theta$.

Let $f_0(v(x))$ denote $f_\theta(v(x))$ for $\theta = 0$, and $f_{\pi/2}(v(x))$ denote $f_\theta(v(x))$ for $\theta = \pi/2$. Then we have the following theorem:

**Theorem 1.** *Assume group representation condition 1 and isotropic scaling condition 2. At any fixed $\boldsymbol{x}$, for the local motion $\delta\boldsymbol{x} = (\delta r \cos\theta, \delta r \sin\theta)$ around $\boldsymbol{x}$, let $\delta\boldsymbol{v} = \boldsymbol{v}(\boldsymbol{x}+\delta\boldsymbol{x}) - \boldsymbol{v}(\boldsymbol{x})$ be the change of vector and $s = \|f_\theta(\boldsymbol{v}(\boldsymbol{x}))\|$, then we have $\|\delta\boldsymbol{v}\| = s\|\delta\boldsymbol{x}\|$. Moreover,*

$$\delta\boldsymbol{v} = f_\theta(\boldsymbol{v}(\boldsymbol{x}))\delta r + o(\delta r) = f_0(\boldsymbol{v}(\boldsymbol{x}))\delta r \cos\theta + f_{\pi/2}(\boldsymbol{v}(\boldsymbol{x}))\delta r \sin\theta + o(\delta r), \qquad (6)$$

*where $f_0(\boldsymbol{v}(\boldsymbol{x}))$ and $f_{\pi/2}(\boldsymbol{v}(\boldsymbol{x}))$ are two orthogonal basis vectors of equal norm s.*

See Supplementary for a proof and Fig. 1(b) for an illustration. Theorem 1 indicates that the local 2D polar system around self-position $\boldsymbol{x}$ in the 2D physical space is embedded conformally as a 2D polar system around vector $\boldsymbol{v}(\boldsymbol{x})$ in the $d$-dimensional neural space, with a scaling factor $s$ (our analysis is local for any fixed $\boldsymbol{x}$, and $s$ may depend on $\boldsymbol{x}$). Conformal embedding is a generalization of isometric embedding, where the metric can be changed by a scaling factor $s$. If $s$ is globally constant for all $\boldsymbol{x}$, then the intrinsic geometry of the codebook manifold $(\boldsymbol{v}(\boldsymbol{x}), \forall\boldsymbol{x})$ remains Euclidean, i.e., flat.

**Why isotropic scaling and conformal embedding?** The neurons are intrinsically noisy. During path integration, the errors may accumulate in $\boldsymbol{v}$. Moreover, when inferring self-position from visual image, it is possible that $\boldsymbol{v}$ is inferred first with error, and then $\boldsymbol{x}$ is decoded from the inferred $\boldsymbol{v}$. Due to isotropic scaling and conformal embedding, locally we have $\|\delta\boldsymbol{v}\| = s\|\delta\boldsymbol{x}\|$, which guarantees that the $\ell_2$ error in $\boldsymbol{v}$ translates proportionally to the $\ell_2$ error in $\boldsymbol{x}$, so that there will not be adversarial perturbations in $\boldsymbol{v}(\boldsymbol{x})$ that cause excessively big errors in $\boldsymbol{x}$. Specifically, we have the following theorem.

**Theorem 2.** *Assume the general transformation model (Eq. (4)) and the isotropic scaling condition. For any fixed $\boldsymbol{x}$, let $s = \|f_\theta(\boldsymbol{v}(\boldsymbol{x}))\|$, which is independent of $\theta$. Suppose the neurons are noisy: $\boldsymbol{v} = \boldsymbol{v}(\boldsymbol{x}) + \boldsymbol{\varepsilon}$, where $\boldsymbol{\varepsilon} \sim \mathcal{N}(0, \tau^2\boldsymbol{I}_d)$ and $d$ is the dimensionality of $\boldsymbol{v}$. Suppose the agent infers its 2D position $\hat{\boldsymbol{x}}$ from $\boldsymbol{v}$ by $\hat{\boldsymbol{x}} = \arg\min_{\boldsymbol{x}'} \|\boldsymbol{v} - \boldsymbol{v}(\boldsymbol{x}')\|^2$, i.e., $\boldsymbol{v}(\hat{\boldsymbol{x}})$ is the projection of $\boldsymbol{v}$ onto the 2D manifold formed by $(\boldsymbol{v}(\boldsymbol{x}'), \forall\boldsymbol{x}')$. Then we have*

$$\mathbb{E}\|\hat{\boldsymbol{x}} - \boldsymbol{x}\|^2 = 2\tau^2/s^2. \qquad (7)$$

See Supplementary for a proof.

**Connection to continuous attractor neural network (CANN) defined on 2D torus**. The group representation condition and the isotropic scaling condition appear to be satisfied by the CANN models [2, 6, 7, 29, 1] that are typically hand-designed on a 2D torus. See Supplementary for details.

## 3 Linear transformation

After studying the general transformation, we now investigate the linear transformation of $\boldsymbol{v}(\boldsymbol{x})$, for the following reasons. (1) It is the simplest transformation for which we can derive explicit algebraic and geometric results. (2) It enables us to connect the isotropic scaling condition to a special class of hexagon grid patterns. (3) In Section 4, we integrate it with the basis expansion model, which is also linear in $\boldsymbol{v}(\boldsymbol{x})$, via unitary group representation theory.

For finite (non-infinitesimal) self-motion, the linear transformation model is:

$$\boldsymbol{v}(\boldsymbol{x} + \Delta\boldsymbol{x}) = F(\boldsymbol{v}(\boldsymbol{x}), \Delta\boldsymbol{x}) = \boldsymbol{M}(\Delta\boldsymbol{x})\boldsymbol{v}(\boldsymbol{x}), \qquad (8)$$

where $\boldsymbol{M}(\Delta\boldsymbol{x})$ is a matrix. The group representation condition becomes $\boldsymbol{M}(\Delta\boldsymbol{x}_1 + \Delta\boldsymbol{x}_2)\boldsymbol{v}(\boldsymbol{x}) = \boldsymbol{M}(\Delta\boldsymbol{x}_2)\boldsymbol{M}(\Delta\boldsymbol{x}_1)\boldsymbol{v}(\boldsymbol{x})$, i.e., $\boldsymbol{M}(\Delta\boldsymbol{x})$ is a matrix representation of self-motion $\Delta\boldsymbol{x}$, and $\boldsymbol{M}(\Delta\boldsymbol{x})$ acts on the coding manifold $(\boldsymbol{v}(\boldsymbol{x}), \forall\boldsymbol{x})$. For egocentric parametrization of self-motion $(\Delta r, \theta)$, we can further write $\boldsymbol{M}(\Delta\boldsymbol{x}) = \boldsymbol{M}_\theta(\Delta r)$ for $\Delta\boldsymbol{x} = (\Delta r \cos\theta, \Delta r \sin\theta)$, and the linear model becomes $\boldsymbol{v}(\boldsymbol{x} + \Delta\boldsymbol{x}) = F(\boldsymbol{v}(\boldsymbol{x}), \Delta r, \theta) = \boldsymbol{M}_\theta(\Delta r)\boldsymbol{v}(\boldsymbol{x})$.

### 3.1 Algebraic structure: matrix Lie algebra and Lie group

For the linear model (Eq. (8)), the directional derivative is: $f_\theta(\boldsymbol{v}(\boldsymbol{x})) = F'(\boldsymbol{v}(\boldsymbol{x}), 0, \theta) = \boldsymbol{M}'_\theta(0)\boldsymbol{v}(\boldsymbol{x}) = \boldsymbol{B}(\theta)\boldsymbol{v}(\boldsymbol{x})$, where $\boldsymbol{B}(\theta) = \boldsymbol{M}'_\theta(0)$, which is the derivative of $\boldsymbol{M}_\theta(\Delta r)$ with respect to $\Delta r$ at 0. For infinitesimal self-motion, the transformation model in Eq. (5) becomes

$$\boldsymbol{v}(\boldsymbol{x} + \delta\boldsymbol{x}) = (\boldsymbol{I} + \boldsymbol{B}(\theta)\delta r)\boldsymbol{v}(\boldsymbol{x}) + o(\delta r), \qquad (9)$$

where $\boldsymbol{I}$ is the identity matrix. It can be considered a linear recurrent network where $\boldsymbol{B}(\theta)$ is the learnable weight matrix. We have the following theorem for the algebraic structure of the linear transformation.

**Theorem 3.** *Assume the linear transformation model so that for infinitesimal self-motion $(\delta r, \theta)$, the model is in the form of Eq. (9), then for finite displacement $\Delta r$,*

$$v(x + \Delta x) = M_\theta(\Delta r)v(x) = \exp(B(\theta)\Delta r)v(x). \tag{10}$$

*Proof.* We can divide $\Delta r$ into $N$ steps, so that $\delta r = \Delta r/N \to 0$ as $N \to \infty$, and

$$v(x + \Delta x) = (I + B(\theta)(\Delta r/N) + o(1/N))^N v(x) \to \exp(B(\theta)\Delta r)v(x) \tag{11}$$

as $N \to \infty$. The matrix exponential map is defined by $\exp(A) = \sum_{n=0}^{\infty} A^n/n!$. $\square$

The above math underlies the relationship between matrix Lie algebra and matrix Lie group in general [38]. For a fixed $\theta$, the set of $M_\theta(\Delta r) = \exp(B(\theta)\Delta r)$ for $\Delta r \in \mathbb{R}$ forms a *matrix Lie group*, which is both a group and a manifold. The tangent space of $M_\theta(\Delta r)$ at identity $I$ is called *matrix Lie algebra*. $B(\theta)$ is the basis of this tangent space, and is often referred to as the *generator matrix*.

**Path integration**. If the agent starts from $x_0$, and make a sequence of moves $((\Delta r_t, \theta_t), t = 1, ..., T)$, then the vector representation of self-position is updated by

$$v_t = \exp(B(\theta_t)\Delta r_t)v_{t-1}, \tag{12}$$

where $v_0 = v(x_0)$, and $t = 1, ..., T$.

**Approximation to exponential map**. For a finite but small $\Delta r$, $\exp(B(\theta)\Delta r)$ can be approximated by a second-order (or higher-order) Taylor expansion

$$\exp(B(\theta)\Delta r) = I + B(\theta)\Delta r + B(\theta)^2\Delta r^2/2 + o(\Delta r^2). \tag{13}$$

### 3.2 Geometric structure: rotation, periodicity, metic and error correction

If we assume $B(\theta) = -B(\theta)^\top$, i.e., skew-symmetric, then $I + B(\theta)\delta r$ in Eq. (9) is a rotation matrix operating on $v(x)$, due to the fact that $(I + B(\theta)\delta r)(I + B(\theta)\delta r)^\top = I + O(\delta r^2)$. For finite $\Delta r$, $\exp(B(\theta)\Delta r)$ is also a rotation matrix, as it equals to the product of $N$ matrices $I + B(\theta)(\Delta r/N)$ (Eq. (11)). The geometric interpretation is that, if the agent moves along the direction $\theta$ in the physical space, the vector $v(x)$ is rotated by the matrix $B(\theta)$ in the neural space, while the $\ell_2$ norm $\|v(x)\|^2$ remains fixed. We may interpret $\|v(x)\|^2 = \sum_{i=1}^{d} v_i(x)^2$ as the total energy of grid cells. See Fig. 1(b).

The angle of rotation is given by $\|B(\theta)v(x)\|\delta r/\|v(x)\|$, because $\|B(\theta)v(x)\|\delta r$ is the arc length and $\|v(x)\|$ is the radius. If we further assume the isotropic scaling condition, which becomes that $\|f_\theta(v(x))\| = \|B(\theta)v(x)\|$ is constant over $\theta$ for the linear model, then the angle of rotation can be written as $\mu\delta r$, where $\mu = \|B(\theta)v(x)\|/\|v(x)\|$ is independent of $\theta$. Geometrically, $\mu$ tells us how fast the vector rotates in the neural space as the agent moves in the physical space. In practice, $\mu$ can be much bigger than 1 for the learned model, thus the vector can rotate back to itself in a short distance, causing the periodic patterns in the elements of $v(x)$. $\mu$ captures the notion of metric.

For $\mu \gg 1$, the conformal embedding in Fig. 1 (b) **magnifies** the local motion in Fig. 1 (a), and this enables error correction [34]. More specifically, we have the following result, which is based on Theorem 2.

**Proposition 1.** *Assume the linear transformation model (Eq. (9)) and the isotropic scaling condition 2. For any fixed $x$, let $\mu = \|B(\theta)v(x)\|/\|v(x)\|$. Suppose $v = v(x) + \varepsilon$, where $\varepsilon \sim \mathcal{N}(0, \tau^2 I_d)$ and $\tau^2 = \alpha^2(\|v(x)\|^2/d)$, so that $\alpha^2$ measures the variance of noise relative to the average magnitude of $(v_i(x)^2, i = 1, ..., d)$. Suppose the agent infers its 2D position $\hat{x}$ from $v$ by $\hat{x} = \arg\min_{x'} \|v - v(x')\|^2$. Then we have*

$$\mathbb{E}\|\hat{x} - x\|^2 = 2\alpha^2/(\mu^2 d). \tag{14}$$

See Supplementary for a proof. By the above proposition, error correction of grid cells is due to two factors: (1) higher dimensionality $d$ of $v(x)$ for encoding 2D positions $x$, and (2) a magnifying $\mu \gg 1$ (our analysis is local for any fixed $x$, and $\mu$ may depend on $x$).

### 3.3 Hexagon grid patterns formed by mixing Fourier waves

In this subsection, we make connection between the isotropic scaling condition 2 and a special class of hexagon grid patterns created by linearly mixing three Fourier plane waves whose directions are $2\pi/3$ apart. We show such linear mixing satisfies the linear transformation model and the isotropic scaling condition.

**Theorem 4.** *Let $e(x) = (\exp(i\langle a_j, x \rangle), j = 1, 2, 3)^\top$, where $(a_j, j = 1, 2, 3)$ are three 2D vectors of equal norm, and the angle between every pair of them is $2\pi/3$. Let $v(x) = Ue(x)$, where $U$ is an arbitrary unitary matrix. Let $B(\theta) = U^* D(\theta) U$, where $D(\theta) = \mathrm{diag}(i\langle a_j, q(\theta)\rangle, j = 1, 2, 3)$, with $q(\theta) = (\cos\theta, \sin\theta)^\top$. Then $(v(x), B(\theta))$ satisfies the linear transformation model (Eq. (9)) and the isotropic scaling condition 2. Moreover, $B(\theta)$ is skew-symmetric.*

See Supplementary for a proof. We would like to emphasize that the above theorem analyzes a special case solution to our linear transformation model, but our optimization-based learning method **does not assume any superposition of Fourier basis functions** as in the theorem. Our experimental results are learned purely by optimizing a loss function based on the simple assumptions of our model with generic vectors and matrices.

We leave it to future work to theoretically prove that the isotropic scaling condition leads to hexagon grid patterns in either the general transformation model or the linear transformation model. The hexagon grid patterns are not limited to superpositions of three plane waves as in the above theorem.

### 3.4 Modules

Biologically, it is well established that grid cells are organized in discrete modules [4, 37] or blocks. We thus partition the vector $v(x)$ into $K$ blocks, $v(x) = (v_k(x), k = 1, ..., K)$. Correspondingly the generator matrices $B(\theta) = \mathrm{diag}(B_k(\theta), k = 1, ..., K)$ are block diagonal, so that each sub-vector $v_k(x)$ is rotated by a sub-matrix $B_k(\theta)$. For the general transformation model, each sub-vector is transformed by a separate sub-network. By the same argument as in Section 3.2, let $\mu_k = \|B_k v_k(x)\| / \|v_k(x)\|$, then $\mu_k$ is the metric of module $k$.

## 4 Interaction with place cells

### 4.1 Place cells

For each $v(x)$, we need to uniquely decode $x$ globally. This can be accomplished via interaction with place cells. Specifically, each place cell fires when the agent is at a specific position. Let $A(x, x')$ be the response map of the place cell associated with position $x'$. It measures the adjacency between $x$ and $x'$. A commonly used form of $A(x, x')$ is the Gaussian adjacency kernel $A(x, x') = \exp(-\|x - x'\|^2 / (2\sigma^2))$. The set of Gaussian adjacency kernels serve as inputs to our optimization-based method to learn grid cells.

### 4.2 Basis expansion

A popular model that connects place cells and grid cells is the following basis expansion model (or PCA-based model) [13]:

$$A(x, x') = \langle v(x), u(x') \rangle = \sum_{i=1}^{d} u_{i,x'} v_i(x), \quad (15)$$

where $v(x) = (v_i(x), i = 1, ..., d)^\top$, and $u(x') = (u_{i,x'}, i = 1, ..., d)^\top$. Here $(v_i(x), i = 1, ..., d)$ forms a set of $d$ basis functions (which are functions of $x$) for expanding $A(x, x')$ (which is a function of $x$ for each place $x'$), while $u(x')$ is the read-out weight vector for place cell at $x'$ and needs to be learned. See Fig. 2 for an illustration. Experimental results on biological brains have shown that the connections from grid cells to place cells are excitatory [42, 31]. We thus assume that $u_{i,x'} \geq 0$ for all $i$ and $x'$.

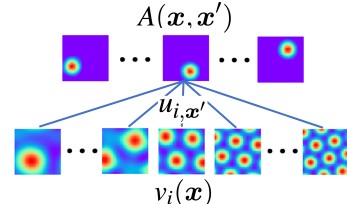

Figure 2: Illustration of basis expansion model $A(x, x') = \sum_{i=1}^{d} u_{i,x'} v_i(x)$, where $v_i(x)$ is the response map of $i$-th grid cell, shown at the bottom, which shows 5 different $i$. $A(x, x')$ is the response map of place cell associated with $x'$, shown at the top, which shows 3 different $x'$. $u_{i,x'}$ is the connection weight.

## 4.3 From group representation to basis functions

The vector representation $v(x)$ generated (or constrained) by the linear transformation model (Eq. (8)) can serve as basis functions of the PCA-based basis expansion model (Eq. (15)), due to the fundamental theorems of Schur [41] and Peter-Weyl [38], which reveal the deep root of Fourier analysis and generalize it to general Lie groups. Specifically, if $M(\Delta x)$ is an irreducible unitary representation of $\Delta x$ that forms a compact Lie group, then the elements $\{M_{ij}(\Delta x)\}$ form a set of orthogonal basis functions of $\Delta x$. Let $v(x) = M(x)v(0)$ (where we choose the origin 0 as the reference point). The elements of $v(x)$, i.e., $(v_i(x), i = 1, ..., d)$, are linear mixings of the basis functions $\{M_{ij}(x)\}$, so that they themselves form a new set of basis functions that serve to expand $(A(x, x'), \forall x')$ that parametrizes the place cells. Thus group representation in our path integration model is a perfect match to the basis expansion model, in the sense that the basis functions are results of group representation.

The basis expansion model (or PCA-based model) (Eq. (15)) assumes that the basis functions are orthogonal, whereas in our work, **we do not make the orthogonality assumption**. Interestingly, the learned transformation model generates basis functions that are close to being orthogonal automatically. See Supplementary for more detailed explanation and experimental results.

## 4.4 Decoding and re-encoding

For a neural response vector $v$, such as $v_t$ in Eq. (12), the response of the place cell associated with location $x'$ is $\langle v, u(x') \rangle$. We can decode the position $\hat{x}$ by examining which place cell has the maximal response, i.e.,

$$\hat{x} = \arg\max_{x'} \langle v, u(x') \rangle. \tag{16}$$

After decoding $\hat{x}$, we can re-encode $v \leftarrow v(\hat{x})$ for error correction. Decoding and re-encoding can also be done by directly projecting $v$ onto the manifold $(v(x), \forall x)$, which gives similar results. See Supplementary for more analysis and experimental results.

## 5 Learning

We learn the model by optimizing a loss function defined based on three model assumptions discussed above: (1) the basis expansion model (Eq. (15)), (2) the linear transformation model (Eq. (10)) and (3) the isotropic scaling condition 2. The input is the set of adjacency kernels $A(x, x'), \forall x, x'$. The unknown parameters to be learned are (1) $(v(x) = (v_k(x), k = 1, ..., K), \forall x)$, (2) $(u(x'), \forall x')$ and (3) $(B(\theta), \forall \theta)$. We assume that there are $K$ modules or blocks and $B(\theta)$ is skew-symmetric, so that $B(\theta)$ are parametrized as block-diagonal matrices $(B_k(\theta), k = 1, ..., K), \forall \theta)$ and only the lower triangle parts of the matrices need to be learned. The loss function is defined as a weighted sum of simple $\ell_2$ loss terms constraining the three model assumptions: $L = L_0 + \lambda_1 L_1 + \lambda_2 L_2$, where

$$L_0 = \mathbb{E}_{x, x'}[A(x, x') - \langle v(x), u(x') \rangle]^2, \text{ (basis expansion)} \tag{17}$$

$$L_1 = \sum_{k=1}^{K} \mathbb{E}_{x, \Delta x} \|v_k(x + \Delta x) - \exp(B_k(\theta)\Delta r)v_k(x)\|^2, \text{ (transformation)} \tag{18}$$

$$L_2 = \sum_{k=1}^{K} \mathbb{E}_{x, \theta, \Delta\theta}[\|B_k(\theta + \Delta\theta)v_k(x)\| - \|B_k(\theta)v_k(x)\|]^2. \text{ (isotropic scaling)} \tag{19}$$

In $L_1$, $\Delta x = (\Delta r \cos\theta, \Delta r \sin\theta)$. $\lambda_1$ and $\lambda_2$ are chosen so that the three loss terms are of similar magnitudes. $A(x, x')$ are given as Gaussian adjacency kernels. For regularization, we add a penalty on $\|u(x')\|^2$, and further assume $u(x') \geq 0$ so that the connections from grid cells to place cells are excitatory [42, 31]. However, note that $u(x') \geq 0$ is not necessary for the emergence of hexagon grid patterns as shown in the ablation studies.

Expectations in $L_0$, $L_1$ and $L_2$ are approximated by Monte Carlo samples. $L$ is minimized by *Adam* [25] optimizer. See Supplementary for implementation details.

It is worth noting that, consistent with the experimental observations, we assume individual place field $A(x, x')$ to exhibit a Gaussian shape, rather than a Mexican-hat pattern (with balanced excitatory center and inhibitory surround) as assumed in previous basis expansion models [13, 33] of grid cells.

**ReLU non-linearity**. We also experiment with a non-linear transformation model where a ReLU activation is added. See Supplementary for details.

## 6 Experiments

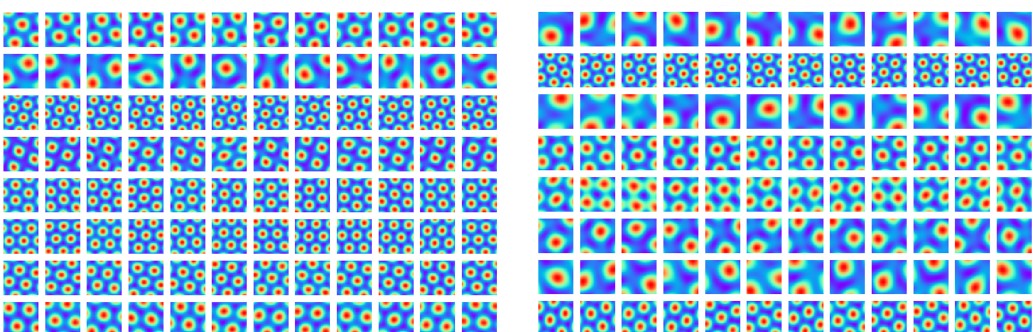

Figure 3: Hexagonal grid firing patterns emerge in the learned network. Every response map shows the firing pattern of one neuron (i.e, one element of $v$) in the 2D environment. Every row shows the firing patterns of the neurons within the same block or module.

We conduct numerical experiments to learn the representations as described in Section 5. Specifically, we use a square environment with size 1m × 1m, which is discretized into a 40 × 40 lattice. For direction, we discretize the circle $[0, 2\pi]$ into 144 directions and use nearest neighbor linear interpolations for values in between. We use the second-order Taylor expansion (Eq. (13)) to approximate the exponential map $\exp(B(\theta)\Delta r)$. The displacement $\Delta r$ are sampled within a small range, i.e., $\Delta r$ is smaller than 3 grids on the lattice. For $A(x, x')$, we use a Gaussian adjacency kernel with $\sigma = 0.07$. $v(x)$ is of $d = 192$ dimensions, which is partitioned into $K = 16$ modules, each of which has 12 cells.

### 6.1 Hexagon grid patterns

Fig. 3 shows the learned firing patterns of $v(x) = (v_i(x), i = 1, ..., d)$ over the 40 × 40 lattice of $x$. Every row shows the learned units belonging to the same block or module. Regular hexagon grid patterns emerge. Within each block or module, the scales and orientations are roughly the same, but with different phases or spatial shifts. For the learned $B(\theta)$, each element shows regular sine/cosine tuning over $\theta$. See Supplementary for more learned patterns.

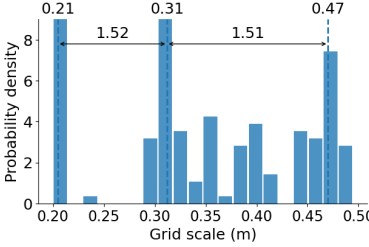

Figure 4: Multi-modal distribution of grid scales of the learned model grid cells. The scale ratios closely match the real data [37].

Table 1: Summary of gridness scores of the patterns learned from different models. To determine valid grid cells, we apply the same threshold of gridness score as in [3], i.e., gridness score > 0.37. For our model, we run 5 trials and report the average and standard deviation.

| Model | Gridness score (↑) | % of grid cells |
|---|---|---|
| [3] (LSTM) | 0.18 | 25.20 |
| [33] (RNN) | 0.48 | 56.10 |
| Ours | **0.90** ± 0.044 | **73.10** ± 1.33 |

We further investigate the characteristics of the learned firing patterns of $v(x)$ using measures adopted from the literature of grid cells. Specifically, the hexagonal regularity, scale and orientation of grid-like patterns are quantified using the gridness score, grid scale and grid orientation [26, 32], which are determined by taking a circular sample of the autocorrelogram of the response map. Table 1 summarizes the results of gridness scores and comparisons with other optimization-based approaches [3, 33]. We apply the same threshold to determine whether a learned neuron can be considered a grid cell as in [3] (i.e., gridness score > 0.37). For our model, 73.10% of the learned neurons exhibit significant hexagonal periodicity in terms of the gridness score. Fig. 4 shows the

histogram of grid scales of the learned grid cell neurons (mean 0.33, range 0.21 to 0.49), which follows a multi-modal distribution. The ratio between neighboring modes are roughly 1.52 and 1.51, which closely matches the theoretical predictions [39, 36] and also the empirical results from rodent grid cells [37]. Collectively, these results reveal striking, quantitative correspondence between the properties of our model neurons and those of the grid cells in the brain.

**Connection to continuous attractor neural network (CANN) defined on 2D torus**. The fact that the learned response maps of each module are shifted versions of a common hexagon periodic pattern implies that the learned codebook manifold forms a 2D torus, and as the agent moves, the responses of the grid cells undergo a cyclic permutation. This is consistent with the CANN models hand-crafted on 2D torus. See Supplementary for a detailed discussion.

**Ablation studies**. We conduct ablation studies to examine whether certain model assumptions are empirically important for the emergence of hexagon grid patterns. The conclusions are highlighted as follows: (1) The loss term $L_2$ (Eq. (19)) constraining the isotropic scaling condition is necessary for learning hexagon grid patterns. (2) The constraint $u(x') \geq 0$ is not necessary for learning hexagon patterns, but the activations can be either excitatory or inhibitory without the constraint. (3) The skew-symmetric assumption on $B(\theta)$ is not important for learning hexagon grid pattern. (4) Hexagon patterns always emerge regardless of the choice of block size and number of blocks. (5) Multiple blocks or modules are necessary for the emergence of hexagon grid patterns of multiple scales. See Fig. 5 for several learned patterns and Supplementary for the full studies.

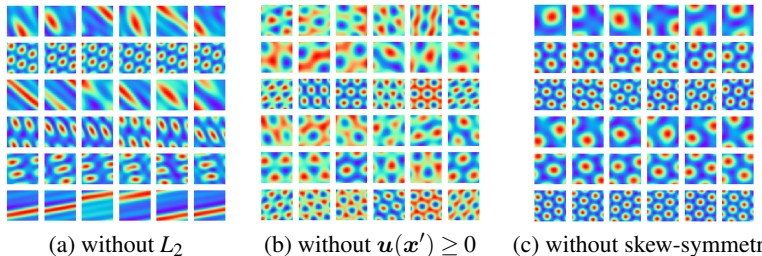

    (a) without $L_2$         (b) without $u(x') \geq 0$    (c) without skew-symmetry

Figure 5: Learned response maps in ablation studies where a certain model assumption is removed. (a) Remove the loss term $L_2$. (b) Remove the assumption $u(x') \geq 0$. (c) Remove the skew-symmetric assumption on $B(\theta)$.

## 6.2 Path integration

We then examine the ability of the learned model on performing multi-step path integration, which can be accomplished by recurrently updating $v_t$ (Eq. (12)) and decoding $v_t$ to $x_t$ for $t = 1, ..., T$ (Eq. (16)). Re-encoding $v_t \leftarrow v(x_t)$ after decoding is adopted. Fig. 6(a) shows an example trajectory of accurate path integration for number of time steps $T = 30$. As shown in Fig. 6(b), with re-encoding, the path integration error remains close to zero over a duration of 500 time steps ($< 0.01$ cm, averaged over 1,000 episodes), even if the model is trained with the single-time-step transformation model (Eq. (18)). Without re-encoding, the error goes slight higher but still remains small (ranging from 0.0 to 4.2 cm, mean 1.9 cm in the 1m $\times$ 1m environment). Fig. 6(c) summarizes the path integration performance by fixing the number of blocks and altering the block size. The performance of path integration would be improved as the block size becomes larger, i.e., with more neurons in each module. When block size is larger than 16, path integration is very accurate for the time steps tested.

**Error correction**. See Supplementary for numerical experiments on error correction, which show that the learn model is still capable of path integration when we apply Gaussian white noise errors or Bernoulli drop-out errors to $v_t$.

## 6.3 Additional experiments on path planning and egocentric vision

We also conduct additional experiments on path planning and egocentric vision with our model. Path planning can be accomplished by steepest ascent on the adjacency to the target position. For egocentric vision, we learn an extra generator network that generates the visual image given the position encoding formed by the grid cells. See Supplementary for details.

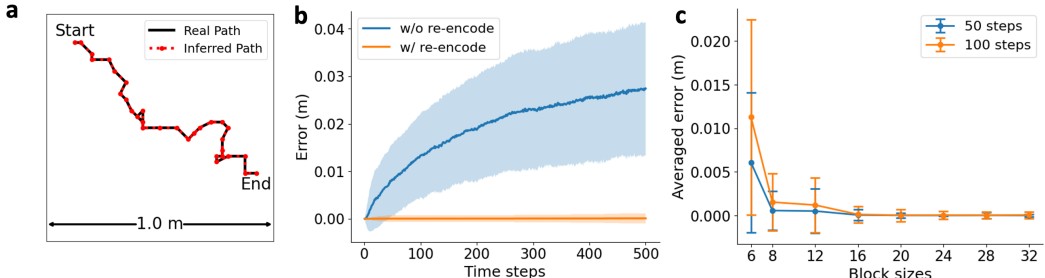

Figure 6: The learned model can perform accurate path integration. (a) Black: example trajectory. Red: inferred trajectory. (b) Path integration error over number of time steps, for procedures with re-encoding and without re-encoding. (c) Path integration error with fixed number of blocks and different block sizes, for 50 and 100 time steps. The error band in (b) and error bar in (c) are standard deviations computed over 1,000 episodes.

## 7   Related work

Our work is related to several lines of previous research on modeling grid cells. First, RNN models have been used to model grid cells and path integration. The traditional approach uses simulation-based models with hand-crafted connectivity, known as continuous attractor neural network (CANN) [2, 6, 7, 29, 1]. On the other hand, more recently two pioneering papers [9, 3] developed optimization-based RNN approaches to learn the path integration model and discovered that grid-like response patterns can emerge in the optimized networks. These results are further substantiated in [33, 8]. Our work analyzes the properties of the general recurrent model for path integration, and these properties seem to be satisfied by the hand-crafted CANN models. Our method belongs to the scheme of optimization-based approaches, and the learned response maps share similar properties as assumed by the CANN models.

Second, our work differs from the PCA-based basis expansion models [13, 33, 35] in that, unlike PCA, we make no assumption about the orthogonality between the basis functions, and the basis functions are generated by the transformation model. Furthermore, in previous basis expansion models [13, 33], place fields with Mexican-hat patterns (with balanced excitatory center and inhibitory surround) had to be assumed in order to obtain hexagonal grid firing patterns. However, experimentally measured place fields in biological brains were instead well characterized by Gaussian functions. Crucially, in our model, hexagonal grids emerge from learning with Gaussian place fields, and there is no need to assume any additional surround mechanisms or difference of Gaussians kernels.

In another related paper, [19] proposed matrix representation of 2D self-motion, while our work analyzes general transformations. Our investigation of the special case of linear transformation model reveals the matrix Lie group and the matrix Lie algebra of rotation group. Our work also connects the linear transformation model to the basis expansion model via unitary group representation theory.

## 8   Conclusion

This paper analyzes the recurrent model for path integration calculations by grid cells. We identify a group representation condition and an isotropic scaling condition that give rise to locally conformal embedding of the self-motion. We study a linear prototype model that reveals the matrix Lie group of rotation, and explore the connection between the isotropic scaling condition and hexagon grid patterns. In addition to these theoretical investigations, our numerical experiments demonstrate that our model can learn hexagon grid patterns for the response maps of grid cells, and the learned model is capable of accurate long distance path integration.

In this work, the numerical experiments are mostly limited to the linear transformation model, with the exception of an experiment with ReLU non-linearity. We will conduct experiments on the other non-linear transformation models, especially the forms assumed by the hand-crafted continuous attractor neural networks. Moreover, we assume that the agent navigates within a square open-field environment without obstacles or rewards. It is worthwhile to explore more complicated environments, including 3D environment.

## Acknowledgments and Disclosure of Funding

The work was supported by NSF DMS-2015577, ONR MURI project N00014-16-1-2007, DARPA XAI project N66001-17-2-4029, and XSEDE grant ASC170063. We thank Yaxuan Zhu from UCLA Department of Statistics for his help with experiments on egocentric vision. We thank Dr. Wenhao Zhang for sharing his knowledge and insights on continuous attractor neural networks. We thank Sirui Xie for discussions. We thank the three reviewers for their constructive comments.

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
