# Supplementary Materials for "On Path Integration of Grid Cells: Group Representation and Isotropic Scaling"

**Ruiqi Gao**[1*]
ruiqigao@ucla.edu

**Jianwen Xie**[2]
jianwen@ucla.edu

**Xue-Xin Wei**[3]
weixx@utexas.edu

**Song-Chun Zhu**[1,4,5]
sczhu@stat.ucla.edu

**Ying Nian Wu**[1]
ywu@stat.ucla.edu

[1]Department of Statistics, UCLA    [2]Cognitive Computing Lab, Baidu Research
[3]Department of Neuroscience, UT Austin    [4]Department of Computer Science, UCLA
[5]Beijing Institute for General Artificial Intelligence (BIGAI)

## Contents

---

[*]The author is now a Research Scientist at Google Brain team.

35th Conference on Neural Information Processing Systems (NeurIPS 2021).

# 1 Theoretical analysis

## 1.1 Graphical illustrations of key equations

Fig. 1 illustrates key equations in the main text as well as in the supplementary materials.

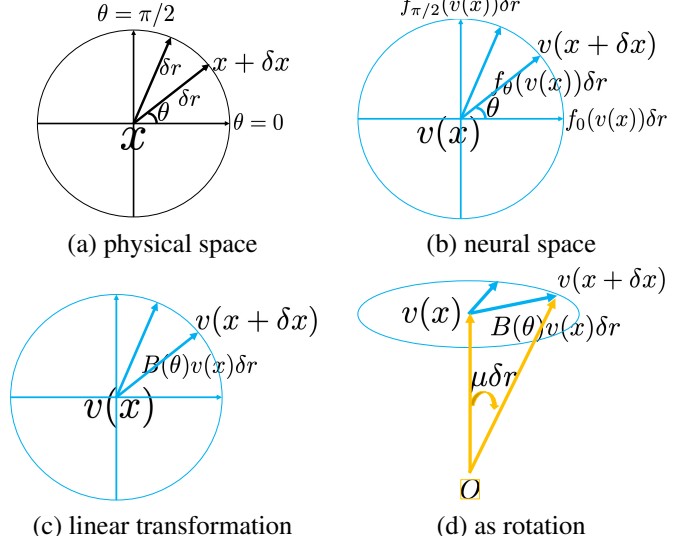

(a) physical space

(b) neural space

(c) linear transformation

(d) as rotation

Figure 1: Color-coded illustration. (a) In the 2D physical space, the agent moves from $\boldsymbol{x}$ to $\boldsymbol{x} + \delta\boldsymbol{x}$, where $\delta\boldsymbol{x} = (\delta r \cos\theta, \delta r \sin\theta)$, i.e., the agent moves by $\delta r$ along the direction $\theta$. We also show a displacement of $\delta r$ in a different direction. (b) In the $d$-dimensional neural space, the vector $\boldsymbol{v}(\boldsymbol{x})$ is changed to $\boldsymbol{v}(\boldsymbol{x} + \delta\boldsymbol{x}) = F(\boldsymbol{v}(\boldsymbol{x}), \delta r, \theta) = \boldsymbol{v}(\boldsymbol{x}) + f_\theta(\boldsymbol{v}(\boldsymbol{x}))\delta r + o(\delta r)$, where the displacement is $f_\theta(\boldsymbol{v}(\boldsymbol{x}))\delta r = f_0(\boldsymbol{v}(\boldsymbol{x}))\delta r \cos\theta + f_{\pi/2}(\boldsymbol{v}(\boldsymbol{x}))\delta r \sin\theta$. Under the isotropic condition that $\|f_\theta(\boldsymbol{v}(\boldsymbol{x}))\|$ is constant over $\theta$, the local 2D self-motion $\delta\boldsymbol{x}$ at $\boldsymbol{x}$ in the 2D physical space is embedded conformally into the neural space as a 2D subspace around $\boldsymbol{v}(\boldsymbol{x})$. (c) Linear transformation, where $f_\theta(\boldsymbol{v}(\boldsymbol{x})) = B(\theta)\boldsymbol{v}(\boldsymbol{x})$. (d) 3D perspective view of linear transformation as a rotation: $\boldsymbol{v}(\boldsymbol{x} + \delta\boldsymbol{x})$ is a rotation of $\boldsymbol{v}(\boldsymbol{x})$, and the angle of rotation is $\mu\delta r$, where $\mu = \|B(\theta)\boldsymbol{v}(\boldsymbol{x})\|/\|\boldsymbol{v}(\boldsymbol{x})\|$ ($\mu$ may depend on $\boldsymbol{x}$).

## 1.2 Proof of Theorem 1 on conformal embedding

*Proof:* See Fig. 1(a) and (b) for an illustration. Consider the self-motion $\delta\boldsymbol{x} = (\delta r \cos\theta, \delta r \sin\theta)$,

$$\boldsymbol{v}(\boldsymbol{x} + \delta\boldsymbol{x}) = F(\boldsymbol{v}(\boldsymbol{x}), \delta r, \theta) = \boldsymbol{v}(\boldsymbol{x}) + f_\theta(\boldsymbol{v}(\boldsymbol{x}))\delta r + o(\delta r). \tag{1}$$

We can decompose the self-motion $\delta\boldsymbol{x}$ into two steps. First move along the direction 0 by $\delta r \cos\theta$, and then move along the direction $\pi/2$ by $\delta r \sin\theta$. Then under the **group representation condition**:

$$\begin{aligned}
\boldsymbol{v}(\boldsymbol{x} + \delta\boldsymbol{x}) &= F[F(\boldsymbol{v}(\boldsymbol{x}), \delta r \cos\theta, 0), \delta r \sin\theta, \pi/2)] \\
&= F[\boldsymbol{v}(\boldsymbol{x}) + f_0(\boldsymbol{v}(\boldsymbol{x}))\delta r \cos\theta + o(\delta r), \delta r \sin\theta, \pi/2] \\
&= [\boldsymbol{v}(\boldsymbol{x}) + f_0(\boldsymbol{v}(\boldsymbol{x}))\delta r \cos\theta] + f_{\pi/2}[\boldsymbol{v}(\boldsymbol{x}) + f_0(\boldsymbol{v}(\boldsymbol{x}))\delta r \cos\theta + o(\delta r)]\delta r \sin\theta + o(\delta r) \\
&= \boldsymbol{v}(\boldsymbol{x}) + f_0(\boldsymbol{v}(\boldsymbol{x}))\delta r \cos\theta + f_{\pi/2}(\boldsymbol{v}(\boldsymbol{x}))\delta r \sin\theta + o(\delta r),
\end{aligned} \tag{2}$$

The last equation holds because assuming the derivative $f'_{\pi/2}(\boldsymbol{v}(\boldsymbol{x}))$ exists, then by first-order Taylor expansion,

$$f_{\pi/2}[\boldsymbol{v}(\boldsymbol{x}) + f_0(\boldsymbol{v}(\boldsymbol{x}))\delta r \cos\theta + o(\delta r)]\delta r \sin\theta \tag{3}$$

$$= [f_{\pi/2}(\boldsymbol{v}(\boldsymbol{x})) + f'_{\pi/2}(\boldsymbol{v}(\boldsymbol{x}))f_0(\boldsymbol{v}(\boldsymbol{x}))\delta r \cos\theta + o(\delta r)]\delta r \sin\theta \tag{4}$$

$$= f_{\pi/2}(\boldsymbol{v}(\boldsymbol{x}))\delta r \sin\theta + o(\delta r). \tag{5}$$

Since $\boldsymbol{v}(\boldsymbol{x} + \delta\boldsymbol{x}) = \boldsymbol{v}(\boldsymbol{x}) + f_\theta(\boldsymbol{v}(\boldsymbol{x}))\delta r + o(\delta r)$, by Eq. (2) we have $f_\theta(\boldsymbol{v}(\boldsymbol{x})) = f_0(\boldsymbol{v}(\boldsymbol{x}))\cos\theta + f_{\pi/2}(\boldsymbol{v}(\boldsymbol{x}))\sin\theta$, which is a 2D basis expansion. We are yet to prove that the two basis vectors $f_0(\boldsymbol{v}(\boldsymbol{x}))$ and $f_{\pi/2}(\boldsymbol{v}(\boldsymbol{x}))$ are orthogonal with equal norm.

For notational simplicity, let $v_1 = f_0(v(x))$ and $v_2 = f_{\pi/2}(v(x))$. Then under the **isotropic scaling condition**, $\|v_1\| = \|v_2\| = \|f_\theta(v(x))\| = s$, and $f_\theta(v(x)) = v_1\cos\theta + v_2\sin\theta$ for any $\theta$. Then we have that for any $\theta$,

$$s^2 = \|f_\theta(v(x))\|^2 = \|v_1\cos\theta + v_2\sin\theta\|^2 = s^2 + 2\langle v_1, v_2\rangle\cos\theta\sin\theta. \tag{6}$$

Thus $\langle v_1, v_2\rangle = 0$, i.e., $f_0(v(x)) \perp f_{\pi/2}(v(x))$. This leads to the conformal embedding of the local 2D polar system in the physical space as a 2D polar system in the $d$-dimensional neural space, with a scaling factor $s$ (which may depend on $x$). $\square$

## 1.3 Proofs of Theorem 2 and Proposition 1 on error correction

*Proof of Theorem 2:* By Theorem 1, for a fixed self-position $x$, we embed the 2D local neighborhood around $x$ as a local 2D plane around $v(x)$ in the $d$-dimensional neural space. A local perturbation in self-position, $\delta x$, is translated into a local perturbation in $v(v + \delta x)$, so that

$$\|\delta v\|^2 = \|f_\theta(v(x))\delta r + o(\delta r)\|^2 = s^2\|\delta x\|^2, \tag{7}$$

where $\delta v = v(x + \delta x) - v(x)$.

Suppose the agent infers its 2D position $\hat{x}$ by $\hat{x} = \arg\min_{x'}\|v - v(x')\|^2$, which amounts to projecting $v$ onto the local 2D plane around $v(x)$. The projected vector $v(\hat{x})$ on the local 2D plane is $v(x) + \delta v$, where $\delta v$ is the projection of $\varepsilon$ onto the 2D plane. More specifically, let $(v_1, v_2)$ be an orthonormal basis of the local 2D plane centered at $v(x)$. Then $\delta v$ can be written as $e_1 v_1 + e_2 v_2$, where

$$e = (e_1, e_2)^\top = (v_1, v_2)^\top\varepsilon \sim \mathcal{N}(0, \tau^2 I_2). \tag{8}$$

Let $\delta x = \hat{x} - x$. Due to **isotropic scaling and conformal embedding**, the $\ell_2$ squared error translate according to

$$\|\delta x\|^2 = \|\delta v\|^2/s^2 = (e_1^2 + e_2^2)/s^2, \tag{9}$$

whose expectation is $2\tau^2/s^2$. Thus $\mathbb{E}\|\hat{x} - x\|^2 = 2\tau^2/s^2$. $\square$

*Proof of Proposition 1:* It is reasonable to assume $\tau^2 = \alpha^2(\|v(x)\|^2/d)$, where $\alpha^2$ measures the variance of noise relative to $\|v(x)\|^2/d$, which is the average of $(v_i(x)^2, i = 1, ..., d)$. In other words, $\alpha^2$ measures the noise level.

In the linear case, the metric is

$$\mu = \|f_\theta(v(x))\|/\|v(x)\| = \|B(\theta)v(x)\|/\|v(x)\| = s/\|v(x)\|, \tag{10}$$

which measures how fast $v(x)$ rotates in the neural space as $x$ changes. Then

$$\mathbb{E}\|\delta x\|^2 = 2\alpha^2/(\mu^2 d). \tag{11}$$

The above scaling shows that error correction depends on two factors. One is the metric $\mu$, and the other is the dimensionality $d$, i.e., the number of neurons. These correspond to two phases of error correction. One is to project the $d$-dimensional $\varepsilon$ to the 2-dimensional $\delta v$. The bigger $d$ is, the better the error correction. The other is to translate $\|\delta v\|^2$ to $\|\delta x\|^2$. The bigger $\mu$ is, the better the error correction. $\square$

## 1.4 Proof of Theorem 4 on hexagon grid patterns

*Proof:* Let $e(x) = (\exp(i\langle a_j, x\rangle), j = 1, 2, 3)^\top$, where $(a_j, j = 1, 2, 3)$ are three 2D vectors of equal norm, and the angle between every pair of them is $2\pi/3$. Let $v(x) = Ue(x)$, where $U$ is an arbitrary unitary matrix, i.e., $U^*U = I$. Then $\|v(x)\|^2 = \|e(x)\|^2 = 3, \forall x$, and $e(x) = U^*v(x)$. For self-motion $\delta x = (\delta r\cos\theta, \delta r\sin\theta) = q(\theta)\delta r$, let

$$\begin{aligned}
\Lambda(\delta x, \theta) &= \mathrm{diag}(\exp(\langle a_j, \delta x\rangle), j = 1, 2, 3) \\
&= \mathrm{diag}(\exp(\langle a_j, q(\theta)\rangle\delta r), j = 1, 2, 3) \\
&= I + \mathrm{diag}(i\langle a_j, q(\theta)\rangle), j = 1, 2, 3)\delta r + o(\delta r) \\
&= I + D(\theta)\delta r + o(\delta r).
\end{aligned} \tag{12}$$

Then

$$
\begin{aligned}
v(x + \delta x) &= Ue(x + \delta x) \\
&= U\Lambda(\delta x, \theta)e(x) \\
&= U\Lambda(\delta x, \theta)U^*v(x) \\
&= (I + UD(\theta)U^*v(x)\delta r)v(x) + o(\delta r) \\
&= (I + B(\theta)\delta r)v(x) + o(\delta r),
\end{aligned}
\tag{13}
$$

where $B(\theta) = UD(\theta)U^*$, and $B(\theta) = -B(\theta)^*$. For isotropic condition,

$$
\begin{aligned}
\|B(\theta)v(x)\|^2 &= \|D(\theta)e(x)\|^2 \\
&= \sum_{j=1}^{3} \langle a_j, q(\theta) \rangle^2 \\
&= \text{const}\|a_j\|^2\|q(\theta)\|^2 = \text{const}\|a_j\|^2,
\end{aligned}
\tag{14}
$$

which is independent of $\theta$, because $(a_j, j = 1, 2, 3)$ forms a **tight frame** in 2D.

One example of $U$ is the following matrix:

$$
\frac{1}{\sqrt{3}}
\begin{pmatrix}
1 & 1 & 1 \\
1 & \exp(i2\pi/3) & \exp(-i2\pi/3) \\
1 & \exp(-i2\pi/3) & \exp(i2\pi/3)
\end{pmatrix}
\tag{15}
$$

The resulting $(v_i(x), i = 1, 2, 3)$ have the same orientation but different phases, i.e., they are spatially shifted versions of each other. $\square$

The limitation of Theorem 4 is that we only show $v(x) = Ue(x)$ satisfies the linear model and the isotropic scaling condition, but we did not show that linear model with isotropic condition only has solutions that are hexagon grid patterns.

## 1.5 From group representation to orthogonal basis functions

Group representation is a central theme in modern mathematics and physics. In particular, it leads to a deep understanding and generalization of Fourier analysis or harmonic analysis.

For the set of $(\Delta x)$ that form a group, a matrix representation $M(\Delta x)$ is equivalent to another representation $\tilde{M}(\Delta x)$ if there exists an invertible matrix $P$ such that $\tilde{M}(\Delta x) = PM(\Delta x)P^{-1}$ for each $x$. A matrix representation is reducible if it is equivalent to a block diagonal matrix representation, i.e., we can find a matrix $P$, such that $PM(\Delta x)P^{-1}$ is block diagonal for every $\Delta x$. Suppose the group is a finite group or a compact Lie group, and $M$ is a unitary representation, i.e., $M(\Delta x)$ is a unitary matrix. If $M$ is block-diagonal, $M = \text{diag}(M_k, k = 1, ..., K)$, with non-equivalent blocks, and each block $M_k$ cannot be further reduced, then the matrix elements $(M_{kij}(\Delta x))$ are orthogonal basis functions of $\Delta x$. Such orthogonality relations are proved by Schur [15] for finite group, and by Peter-Weyl for compact Lie group [13]. For our case, theoretically the group of displacements $\Delta x$ in the 2D domain is $\mathbb{R}^2$, but we learn our model within a finite range, and we further discretize the range into a lattice. Thus the above orthogonal relations hold.

In our model, we also assume block diagonal $M$, and we call each block a module. However, we do not assume each module is irreducible, i.e., each module itself may be further diagonalized into a block diagonal matrix of irreducible sub-blocks. Thus the elements within the same module $v_k(x)$ may be linear mixings of orthogonal basis functions of the irreducible sub-blocks, and the linear mixings themselves are not necessarily orthogonal.

Fig. 2 visualizes the correlation between pairs of the learned $v_i(x)$ and $v_j(x)$, $i, j = 1, ..., d$. For different $i$ and $j$, the correlations between different $v_i(x)$ and $v_j(x)$ are close to zero; i.e., they are nearly orthogonal to each other. The average absolute value of correlation is 0.09, and the within-block average value is about the same as the between-block average value.

Unlike previous work on learning basis expansion model (or PCA-based model [6]), we **do not constrain the basis functions $v(x) = (v_i(x), i = 1, ..., d)$ to be orthogonal to each other**. Instead, we constrain them by our path integration model via the loss term $L_1$. Nonetheless, the learned $v_i(x)$ are close to being orthogonal in our experiments.

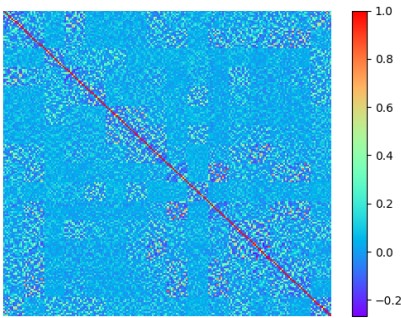

Figure 2: Correlation heatmap for each pair of the learned $v_i(\boldsymbol{x})$ and $v_j(\boldsymbol{x})$. The correlations are computed over $40 \times 40$ lattice of $\boldsymbol{x}$.

## 1.6 Decoding and re-encoding

In the above analysis, the projection of $\boldsymbol{v}$ onto the local 2D plane around $\boldsymbol{v}(\boldsymbol{x})$ is $\hat{\boldsymbol{x}} = \arg\min_{\boldsymbol{x}'} \|\boldsymbol{v} - \boldsymbol{v}(\boldsymbol{x}')\|^2$, which, for the linear model, amounts to decoding $\boldsymbol{v}$ to $\hat{\boldsymbol{x}}$ via

$$\hat{\boldsymbol{x}} = \arg\max_{x'} \langle \boldsymbol{v}, \boldsymbol{v}(\boldsymbol{x}') \rangle, \tag{16}$$

because $\|\boldsymbol{v}(\boldsymbol{x}')\|^2$ is constant. We project $\boldsymbol{v}$ to $\boldsymbol{v}(\hat{\boldsymbol{x}})$, which is an re-encoding of $\boldsymbol{v}$.

We can also perform decoding via the learned $\boldsymbol{u}(\boldsymbol{x}')$:

$$\hat{\boldsymbol{x}} = \arg\max_{x'} \langle \boldsymbol{v}, \boldsymbol{u}(\boldsymbol{x}') \rangle, \tag{17}$$

and re-encoding $\boldsymbol{v} \leftarrow \boldsymbol{v}(\hat{\boldsymbol{x}})$. For the above decoding, the heat map

$$\boldsymbol{h}(\boldsymbol{x}') = \langle \boldsymbol{v}, \boldsymbol{u}(\boldsymbol{x}') \rangle = \langle \boldsymbol{v}(\boldsymbol{x}), \boldsymbol{u}(\boldsymbol{x}') \rangle + \langle \boldsymbol{\varepsilon}, \boldsymbol{u}(\boldsymbol{x}') \rangle = A(\boldsymbol{x}, \boldsymbol{x}') + e(\boldsymbol{x}'), \tag{18}$$

where $e(\boldsymbol{x}') = \langle \boldsymbol{\varepsilon}, \boldsymbol{u}(\boldsymbol{x}') \rangle \sim \mathcal{N}(0, \alpha^2 \|\boldsymbol{v}(\boldsymbol{x})\|^2 \|\boldsymbol{u}(\boldsymbol{x}')\|^2 / d)$. For $A(\boldsymbol{x}, \boldsymbol{x}') = \exp(-\|\boldsymbol{x} - \boldsymbol{x}'\|^2 / (2\sigma^2)) = \langle \boldsymbol{v}(\boldsymbol{x}), \boldsymbol{u}(\boldsymbol{x}') \rangle$, if $\sigma^2$ is small, $A(\boldsymbol{x}, \boldsymbol{x}')$ decreases to 0 quickly, i.e., if $\|\boldsymbol{x}' - \boldsymbol{x}\| > c$, then $A(\boldsymbol{x}, \boldsymbol{x}') < \exp(-c^2 / (2\sigma^2))$, and the chance for the maximum of $\boldsymbol{h}(\boldsymbol{x}')$ to be achieved at an $\boldsymbol{x}'$ so that $\|\boldsymbol{x}' - \boldsymbol{x}\| > c$ can be very small. The above analysis also provides a justification for regularizing $\|\boldsymbol{u}(\boldsymbol{x}')\|^2$ in learning.

For error correction, we want to use small $\sigma^2$. However, for path planning, we need large $\sigma^2$ so that we can assess the adjacency as well as the change of the adjacency between the position on the path and the target position even if they are far apart.

In the experiments in the main text, we use Eq. (17) for decoding. In Fig. 3, we also show the results of path integration using Eq. (16) for decoding, whose performance is even better than Eq. (17). Especially the error would remain 0 over 300 time steps and 1,000 episodes using Eq. (16) with re-encoding. The advantage of (16) is that error correction is achieved within the grid cells system itself without interacting with the place cells.

## 1.7 Connection to continuous attractor neural network (CANN) defined on 2D torus

The CANN models [2, 4, 5, 9, 1] assume that the grid cells $\boldsymbol{v}(\boldsymbol{x}) = (v_i(\boldsymbol{x}), i = 1, ..., d)$ are placed on a finite 2D square lattice with periodic boundary condition, i.e., a 2D torus $\mathbb{T}$. If the lattice is $N \times N$, then $d = N^2$. Let $\boldsymbol{z} \in \mathbb{T}$ be the 2D coordinate of a pixel in $\mathbb{T}$, then each grid cell $v_i$ is placed on a unique $\boldsymbol{z}_i \in \mathbb{T}$.

A CANN model hand-crafts the non-linear recurrent transformation $\boldsymbol{v}(\boldsymbol{x} + \Delta\boldsymbol{x}) = F(\boldsymbol{v}(\boldsymbol{x}), \Delta\boldsymbol{x})$ for some parametric form of $F$, and the coding manifold $(\boldsymbol{v}(\boldsymbol{x}), \forall \boldsymbol{x})$ consists of the attracting fixed points of $F(\cdot, 0)$. In CANN, the recurrent connection weights between a pair of grid cells $(v_i, v_j)$ only depend on the relative positions of the two cells on the 2D torus, $\boldsymbol{z}_i - \boldsymbol{z}_j$, i.e., the connection weights

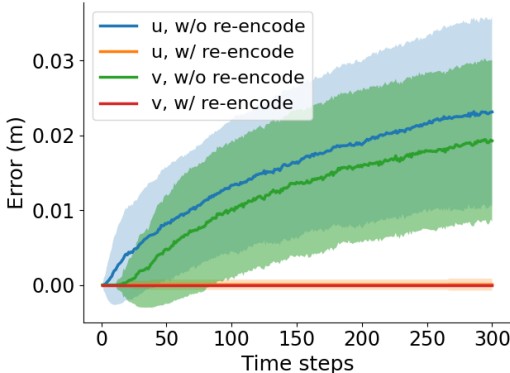

Figure 3: Path integration error over number of time steps. The mean and standard deviation band is computed over 1,000 episodes. "$\boldsymbol{v}$" means decoding by Eq. (16), and "$\boldsymbol{u}$" means decoding by Eq. (17). The squared domain is 1m × 1m.

are convolutional. Such a topographical arrangement may be physically realized on the 2D surface of the cortex in the brain, but it may also be the conceptual interpretation of the connection weights between the grid cells that are not necessarily placed on a physical 2D torus in the brain.

If we place the grid cells $\boldsymbol{v} = (v_i, i = 1, ..., d)$ on the $d = N \times N$ lattice of the 2D torus, either physically or conceptually, then their activities $\boldsymbol{v}(\boldsymbol{x}) = (v_i(\boldsymbol{x}), i = 1, ..., d)$ form an $N \times N$ "image" defined on the 2D torus. The pattern of the "image" may be a localized "bump", i.e., only a local subset of the pixels of the $N \times N$ lattice have non-zero activities. Suppose each self-position $\boldsymbol{x}$ of the agent can be mapped to a "bump" on the 2D torus centered at a corresponding $\boldsymbol{z} \in \mathbb{T}$. When the agent moves in the 2D physical space, i.e., when $\boldsymbol{x}$ changes to $\boldsymbol{x} + \Delta\boldsymbol{x}$, then the "bump" formed by $\boldsymbol{v}(\boldsymbol{x}) = (v_i(\boldsymbol{x}), i = 1, ..., d = N^2)$ moves on the 2D torus from $\boldsymbol{z}$ to $\boldsymbol{z} + \Delta\boldsymbol{z}$, while the shape of the "bump" remains the same. The connection weights of the CANN are hand-crafted so that the recurrent transformation of CANN realizes such a "mirroring" movement of the "bump".

If each displacement $\Delta\boldsymbol{x}$ of the agent in the 2D physical space can be mapped to a displacement $\Delta\boldsymbol{z}$ of the "bump" on the 2D torus $\mathbb{T}$, then the recurrent transformation of the CANN forms a representation of the 2D Euclidean group $\mathbb{R}^2$. If the local movement of the "bump" $\delta\boldsymbol{z}$ on the 2D torus is furthermore conformal to the local movement $\delta\boldsymbol{x}$ of the agent in the 2D physical space, then the local movement of the $d = N^2$ dimensional vector $\boldsymbol{v}(\boldsymbol{x}) = (v_i(\boldsymbol{x}), i = 1, ..., d = N^2)$ formed by the grid cells in the $d$-dimensional neural space, i.e., $\boldsymbol{v}(\boldsymbol{x} + \delta\boldsymbol{x}) - \boldsymbol{v}(\boldsymbol{x})$, is also conformal to the movement $\delta\boldsymbol{x}$ of the agent in the 2D physical space, and the isotropic scaling condition also holds.

In the above understanding, there are three types of movements. (1) The movement $\Delta\boldsymbol{x}$ of the agent in the 2D physical space $\mathbb{R}^2$. (2) The movement $\Delta\boldsymbol{z}$ of the "bump" on the $N \times N$ lattice of 2D torus $\mathbb{T}$. (3) The movement $\boldsymbol{v}(\boldsymbol{x} + \Delta\boldsymbol{x}) - \boldsymbol{v}(\boldsymbol{x})$ in the $d = N^2$-dimensional neural space.

Our model on either the general transformation or the linear transformation does not assume a 2D torus topography. In fact, no 2D topographical structure whatsoever is assumed in our model. The topographical arrangement is not part of our model. Instead, it may be treated as an implementation issue after the model is learned, i.e., how to arrange the grid cells physically on a 2D surface of cortex so that a pair of grid cells with strong connection weights are placed close to each other. It may also be treated as an interpretation issue after the model is learned, i.e., how to interpret the learned connection weights.

Even though our model does not make topographical assumptions, our linear transformation model appears to learn the torus topography automatically. Specifically, in our learned model, the response maps of the grid cells within each module are spatially shifted versions of the same hexagon periodic pattern. Therefore we can identify two directions in the 2D physical space that are $2\pi/3$ apart, so that $\boldsymbol{v}(\boldsymbol{x})$ rotates back to itself as $\boldsymbol{x}$ moves along these two directions for a certain distance. This implies that the codebook manifold $(\boldsymbol{v}(\boldsymbol{x}), \forall \boldsymbol{x})$ forms a 2D torus as assumed by CANN models. Moreover, the fact that the learned response maps of the grid cells within each module are spatially shifted versions of the same hexagon periodic pattern also agrees with the CANN model that moves the "bump" on the 2D torus by "mirroring" the motion in the 2D physical space. The learned hexagon

periodic patterns and the spatial shifts of the response maps may be related to the optimality of the hexagon grid in terms of sampling, interpolation and packing.

Even though the CANN model realizes the movement of the "bump" on the 2D torus by a non-linear recurrent model, such movement is a cyclic permutation of the activities of the grid cells, and the permutation can be realized by a permutation matrix, which is an orthogonal matrix. Thus the $v(x)$ that satisfies the non-linear CANN model also satisfies our linear transformation model, where the linear rotation matrix is a cyclic permutation matrix.

The torus topology is hardly surprising, even for the general transformation model. The Lie group formed by $(F(\cdot, \Delta x), \forall \Delta x)$ is abelian as it is a representation of the 2D additive Euclidean group $\mathbb{R}^2$. If a connected abelian Lie group is compact, then the group is automatically a torus. See [7].

Furthermore, if the scaling factor $s$ is globally a constant for all $x$, then the position embedding $(v(x), \forall x)$ is an isometric embedding up to a global scaling factor, and its intrinsic geometry remains Euclidean. It thus is a **flat torus**.

## 2 Experiments

### 2.1 Implementation details

**Monte Carlo samples.** The expectations in loss terms are approximated by Monte Carlo samples. Here we detail the generation of Monte Carlo samples. For $(x, x')$ used in $L_0 = \mathbb{E}_{x,x'}[A(x,x') - \langle v(x), u(x') \rangle]^2$, $x$ is first sampled uniformly within the entire domain, and then the displacement $dx$ between $x$ and $x'$ is sampled from a normal distribution $\mathcal{N}(0, \sigma^2 I_2)$, where $\sigma = 0.48$. This is to ensure that nearby samples are given more emphasis. We let $x' = x + dx$, and those pairs $(x, x')$ within the range of domain (i.e., 1m × 1m, 40 × 40 lattice) are kept as valid data. For $(x, \Delta x)$ used in $L_1 = \mathbb{E}_{x,\Delta x}|v(x + \Delta x) - \exp(B(\theta)\Delta r)v(x)|^2$, $\Delta x$ is sampled uniformly within a circular domain with radius equal to 3 grids and $(0,0)$ as the center. Specifically, $\Delta r^2$, the squared length of $\Delta x$, is sampled uniformly from $[0,3]$ grids, and $\theta$ is sampled uniformly from $[0, 2\pi]$. We take the square root of the sampled $\Delta r^2$ as $\Delta r$ and let $\Delta x = (\Delta r \cos\theta, \Delta r \sin\theta)$. Then $x$ is uniformly sampled from the region such that both $x$ and $x + \Delta x$ are within the range of domain. For $(\theta, \Delta\theta)$ used in $L_2 = \sum_{k=1}^{K} \mathbb{E}_{x,\theta,\Delta\theta}[\|B_k(\theta + \Delta\theta)v_k(x)\| - \|B_k(\theta)v_k(x)\|]^2$, we uniformly sample $\theta$ and $\theta + \Delta\theta$ from discretized angles, i.e., 144 directions discretized for circle $[0, 2\pi]$. We will study sampling only small $\Delta\theta$ in the future.

**Training details.** The model is trained for 14,000 iterations. At each iteration, the samples are generated online. For the first 8,000 iterations, we update all learnable parameters, while for the following iterations, we fix the learned $v(x)$ and update the other learnable parameters. The initial learning rate is set as 0.003 and is decreased by a factor of 0.5 every 500 iterations after 8,000 iterations. We use Adam [8] optimizer. The model is trained on a single Titan XP GPU. We apply the maximum batch size that can fit into the single GPU, which is 90,000. It takes about 3.5 hours to train the model on a single Titan XP GPU.

**Baseline methods.** In Table 1 of the main text, we compare the learned neurons with the ones from other two optimization-based learning methods [3, 11]. For [3], we run the code released by the authors (`https://github.com/deepmind/grid-cells`) to learn the model and compute gridness scores for the learned neurons. For [11], we use the pre-trained weights released by the authors (`https://github.com/ganguli-lab/grid-pattern-formation`) to get the learned neurons and compute the gridness scores. Both the code of [3] and pre-trained weights of [11] use Apache License V2.

**Usage of data.** In this paper, we mainly use simulated trajectories as training data, and thus we do not think that the data contain any personally identifiable information or offensive content. The only existing data we use is the pre-trained weights of the baseline method [11]. Under Apache License V2, we believe it is fully approved by the authors to use the pre-trained weights.

## 2.2 Learned patterns

Fig. 4 displays the autocorrelograms of learned patterns of $v(x)$.

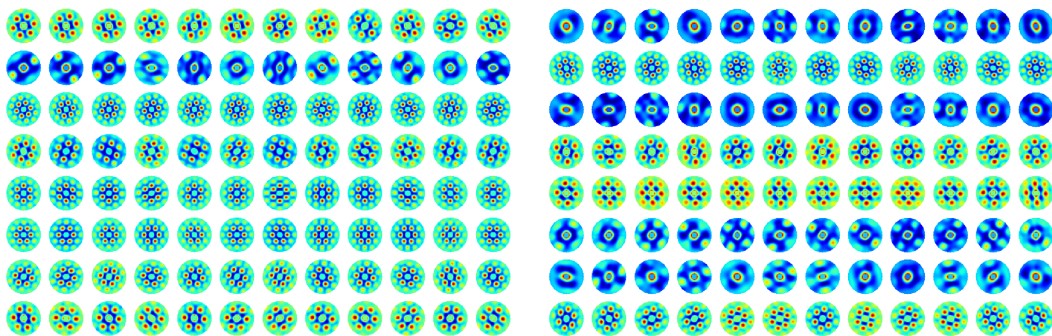

Figure 4: Autocorrelograms of the learned patterns of $v(x)$.

Fig. 5 shows the learned patterns of $u(x)$ with 16 blocks of 12 cells in each block. Regular hexagon patterns also emerge.

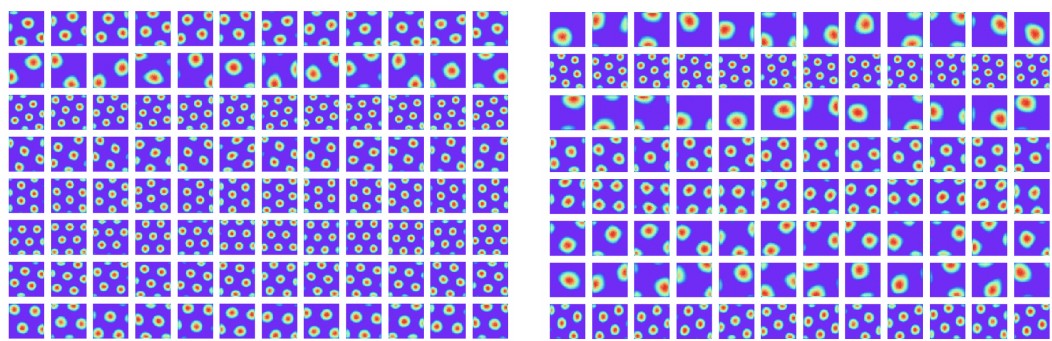

Figure 5: Learned patterns of $u(x)$ with 16 blocks of size 12 cells in each block. Every row shows the learned patterns within the same block.

For learned firing patterns of $v(x)$, we also display the histogram of grid orientations in Fig. 6, where we do not observe clear clusters.

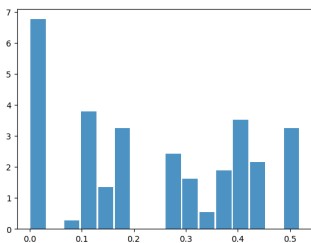

Figure 6: Histogram of grid orientations of the learned firing patterns of $v(x)$.

In Fig. 7, we show the learned patterns of a block of $B(\theta)$. Each element shows significant sine/cosine tuning over $\theta$. For the other blocks, the patterns are all similar.

**Gaussian kernel.** Because $A(x, x')$ is a sharp Gaussian kernel, it contains a whole range of frequencies in the 2D Fourier domain. The learned response maps of the grid cells span a range of frequencies or scales too. Each module or block focuses on a certain frequency band, which corresponds to the metric of the module. We assume individual place field $A(x, x')$ to exhibit a Gaussian shape, rather than a Mexican-hat pattern (with balanced excitatory center and inhibitory

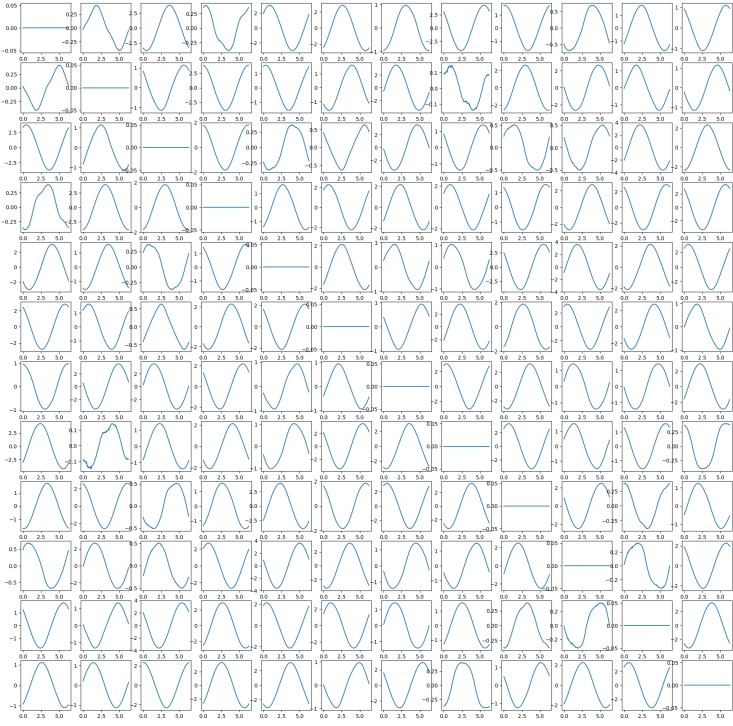

Figure 7: Learned patterns of a block of $\boldsymbol{B}(\theta)$. Each subfigure shows the value of an element in $\boldsymbol{B}(\theta)$ (vertical axis) over $\theta$ (horizontal axis).

surround) as assumed in previous basis expansion models [6, 11] of grid cells. The Mexican-hat or difference of Gaussians pattern occupies a ring in the 2D Fourier domain. It corresponds to a module in our model. But we use isotropic condition to enforce each module to be within a ring in the Fourier domain, and we use different modules to pave the whole Fourier domain.

## 2.3 Error correction

We begin by assessing the ability of error correction of the learned system following the setting in Proposition 1. Specifically, for a given location $\boldsymbol{x}$, suppose the neurons are perturbed by Gaussian noise: $\boldsymbol{v} = \boldsymbol{v}(x) + \boldsymbol{\varepsilon}$, where $\boldsymbol{\varepsilon} \sim \mathcal{N}(0, \tau^2 \boldsymbol{I}_d)$ and $\tau^2 = \alpha^2(\|\boldsymbol{v}(\boldsymbol{x})\|^2/d)$, so that $\alpha^2$ measures the variance of noise relative to the average magnitude of $(v_i(\boldsymbol{x})^2, i = 1, ..., d)$ and $\alpha$ measures the relative standard deviation. We infer the 2D position $\hat{\boldsymbol{x}}$ from $\boldsymbol{v}$ by $\hat{\boldsymbol{x}} = \arg\min_{\boldsymbol{x}'} \|\boldsymbol{v} - \boldsymbol{v}(\boldsymbol{x}')\|^2$. Fig. 8 displays the inference error over the relative standard deviation $\alpha$ of the added Gaussian noise. We also show the results using the learned $\boldsymbol{u}(\boldsymbol{x}')$ for inference (Eq. (17)). The system works remarkably well even if $\alpha = 2$.

We further assess the ability of error correction in long distance path integration. Specifically, along the way of path integration, at every time step $t$, two types of errors are introduced to $\boldsymbol{v}_t$: (1) Gaussian noise or (2) dropout masks, i.e., certain percentage of units are randomly set to zero. Fig. 9 summarizes the path integration performance with different levels of injected errors for $T = 100$, using $\boldsymbol{v}(\boldsymbol{x}')$ (Eq. (16)) or $\boldsymbol{u}(\boldsymbol{x}')$ (Eq. (17)) for decoding. The results show that re-encoding at each step helps error correction, especially for dropout masks. For Gaussian noise, even without decoding and re-encoding at each step, decoding at the final step alone is capable of removing much of the noise. Notably, with re-encoding, the path integration works well even if Gaussian noise with $\alpha = 1$ is added or 50% units are randomly dropped out at each step, indicating that the learned system is robust to different sources of errors.

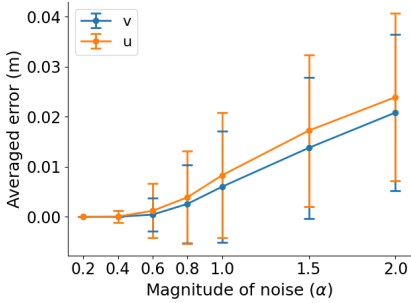

Figure 8: Error correction results following the setting in Proposition 1. The error bar stands for the standard deviation over 1,000 trials. "$\boldsymbol{v}$" means decoding by Eq. (16), and "$\boldsymbol{u}$" means decoding by Eq. (17). The squared domain is 1m × 1m.

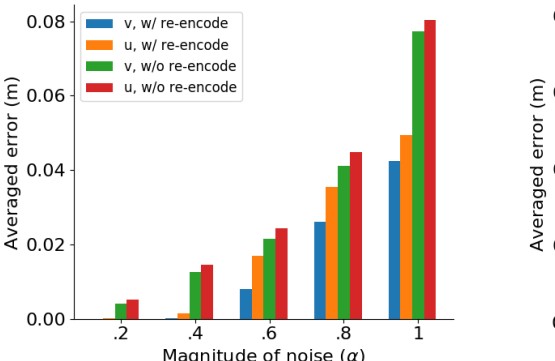 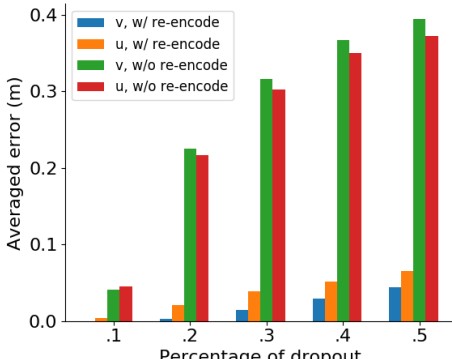

Figure 9: Path integration results with different levels of injected errors. *Left*: Gaussian noise. The magnitude of noise is measured using the average of the squared magnitudes of the units in $\boldsymbol{v}(\boldsymbol{x})$ as the reference. *Right*: dropout masks. Certain percentage of units are randomly set to zero at each step. "$\boldsymbol{v}$" means decoding by Eq. (16), and "$\boldsymbol{u}$" means decoding by Eq. (17). The squared domain is 1m × 1m.

## 2.4   Non-linear transformation model

We test our method with a non-linear transformation model:

$$F(\boldsymbol{v}(\boldsymbol{x}), \Delta r, \boldsymbol{\theta}) = \text{ReLU}(\exp(\boldsymbol{B}(\boldsymbol{\theta})\Delta r)\boldsymbol{v}(\boldsymbol{x})), \tag{19}$$

where we insert $\text{ReLU}(a) = \max(0, a)$ into the linear transformation model.

We use numerical differentiation to define directional derivative

$$f_{\boldsymbol{\theta}}(\boldsymbol{v}(\boldsymbol{x})) = [\boldsymbol{v}(\boldsymbol{x} + \delta\boldsymbol{x}) - \boldsymbol{v}(\boldsymbol{x})]/\delta r, \tag{20}$$

where $\delta\boldsymbol{x} = (\delta r \cos\theta, \delta r \sin\theta)$, with pre-defined $\delta r$. The reason for numerical differentiation is because the derivative of ReLU is an indicator function, which is not differentiable. $f_{\boldsymbol{\theta}}(\boldsymbol{v}(\boldsymbol{x}))$ needs to be differentiable for minimizing the loss function (an alternative to numerical differentiation is to use sigmoid function to approximate the indicator function).

We continue to use the same loss function except with the above two changes. Interestingly, regular hexagon patterns continue to emerge (average gridness score 0.83, percentage of grid cells 70.21%). See Fig. 10 for the learned patterns of $\boldsymbol{v}(\boldsymbol{x})$.

## 2.5   Path planning

Our grid cells model can be applied to path planning. Specifically, according to [12], the adjacency kernel can be modeled by

$$A_{\gamma}(\boldsymbol{x}, \boldsymbol{x}') = \mathbb{E}\left[\sum_{t=0}^{\infty} \gamma^t \mathbb{1}(\boldsymbol{x}_t = \boldsymbol{x}')|\boldsymbol{x}_0 = \boldsymbol{x}\right] = \langle \boldsymbol{v}(\boldsymbol{x}), \boldsymbol{u}_{\gamma}(\boldsymbol{x}')\rangle, \tag{21}$$

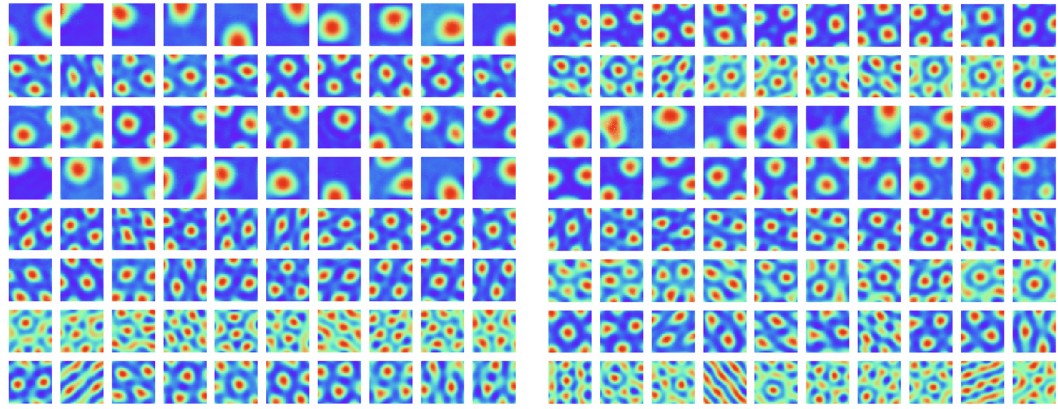

Figure 10: Learned patterns of $v(x)$ with the non-linear transformation model (Eq. (19)). Every row shows the learned patterns within the same block.

where $\gamma$ is the discount factor that controls the temporal and spatial scales, $\mathbb{E}$ is with respect to a random walk exploration policy, and $1(\cdot)$ is the indicator function. For random walk in open field, $A_\gamma(x, x') \propto \exp(-\|x - x'\|^2 / 2\sigma_\gamma^2)$, where $\sigma_\gamma^2$ depends on $\gamma$.

To enable path planning, we need kernels of both big and small spatial scales to account for long and short distance planning respectively. To this end, we discretize $\gamma$ into a finite list of scales, and learn a list of corresponding $u_\gamma(x')$ together with $v(x)$ and $B(\theta)$ using the loss function in Section 5 of the main text.

With the learned model, path planning can be accomplished by steepest ascent on the adjacency to the target position. Specifically, let $\hat{x}$ be the target or destination. Let $x^{(t)}$ be the current position in the path planning process, encoded by $v(x^{(t)})$. The agent plans the next displacement by steepest ascent on

$$A_\gamma(x^{(t)} + \Delta x, \hat{x}) = \langle v(x^{(t)} + \Delta x), u_\gamma(\hat{x}) \rangle = \langle M(\Delta x)v(x^{(t)}), u_\gamma(\hat{x}) \rangle, \tag{22}$$

over allowed $\Delta x$ within a single step, where $M(\Delta x) = \exp(B(\theta)\Delta r)$, with $\Delta x = (\Delta r \cos\theta, \Delta r \sin\theta)$. We plan

$$\Delta x^{(t+1)} = \arg\max_{\Delta x} A_\gamma(x^{(t)} + \Delta x, \hat{x}), \tag{23}$$

and let $x^{(t+1)} = x^{(t)} + \Delta x^{(t+1)}$.

The scale $\gamma$ is selected as the smallest one that satisfies $\max_{\Delta x} \langle M(\Delta x)v(x^{(t)}), u_\gamma(\hat{x}) \rangle > .2$. We can also use $\max_\gamma \max_{\Delta x} \langle M(\Delta x)v(x^{(t)}), u_\gamma(\hat{x}) \rangle$ for scale selection.

We test path planning in the open field environment. The model is first learned using a single-scale kernel function $A_\gamma(x, x') = \exp(-\|x - x'\|^2 / 2\sigma_\gamma^2)$ where $\sigma_\gamma = 0.07$. Then we assume a list of three scales: $\sigma_\gamma = [0.07, 0.14, 0.28]$ and learn the corresponding list of $u_\gamma(x')$. The pool of allowed displacements for a single step is defined as: $dr$ can be 1 or 2 grids, while $\theta$ can be chosen from 200 discretized angles over $[0, 2\pi]$. Fig. 11 demonstrates several examples of path planning in the open field environment, where the agent is able to plan straight path to the target. When $x^{(t)}$ is far from the target, kernel with large $\sigma_\gamma$ is chosen, and as $x^{(t)}$ approaches the target, the chosen kernel gradually switches to the one with small $\sigma_\gamma$. A planning episode is treated as a success if the distance between $x^{(t)}$ and target is smaller than 0.5 grid within 40 time steps. The agent achieves a success rate of 100% (tested for 10,000 episodes).

For a field with obstacles or rewards, we can learn the deformed $A_\gamma(x, x')$ and $(v(x), u_\gamma(x'))$ by temporal difference learning with a random walk exploration policy as suggested in [12]. After learning $A_\gamma(x, x')$ and $(v(x), u_\gamma(x'))$, we can continue to use Eq. (23) for path planning. We shall further study it in future work.

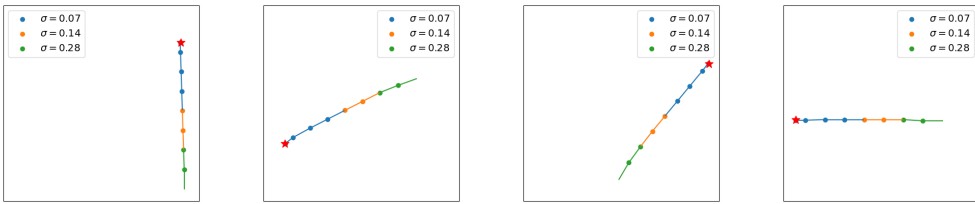

Figure 11: Examples of path planning results in an open field environment. The target is shown as a red star.

## 2.6 Integrating egocentric vision

When the agent moves in darkness, it can infer its self-position by integrating self-motion, as illustrated by our experiments on path integration. If there is visual input, the agent can infer its self-position (as well as head direction) from the visual image alone. We extend our grid cells model to study this problem of egocentric vision, which is important in computer vision.

Specifically, suppose the agent navigates in a 3D scene such as a room, and the height of the eye (or camera) remains fixed. Suppose at 2D self-position $x$ and with head direction $\theta$, the agent sees an image $I$, which is called a posed image. We use the vector representation $v(x)$ in our original grid cells model to represent the 2D self-position $x$, and use another vector representation $h(\theta)$ to represent the head direction $\theta$. If the agent changes its head direction from $\theta$ to $\theta + \Delta\theta$, $h(\theta)$ is transformed to

$$h(\theta + \Delta\theta) = \exp(C\Delta\theta)h(\theta). \tag{24}$$

We assume that there are $K$ modules or blocks in $h(\theta)$ and $C$ is skew-symmetric. This is similar to the transformation of $v(x)$ in our grid cells model.

$(x, \theta)$ is called the pose of the camera (or eye), and we call $(v(x), h(\theta))$ the pose embedding.

To associate the pose embedding $(v(x), h(\theta))$ with the posed image $I$, we use a vector representation or scene embedding $s$ to represent the 3D scene which is shared across different posed images of the same scene, and we learn a generator network $G_\beta$ that maps the embeddings $s$ and $(v(x), h(\theta))$ to the posed image $I$:

$$I = G_\beta(s, v(x), h(\theta)) + \varepsilon, \tag{25}$$

where the generator $G_\beta$ is parametrized by a multi-layer deconvolutional neural network with parameters $\beta$, and $\varepsilon$ is the residual error.

Given the above assumptions, we introduce two extra loss terms in addition to the loss function described in Section 5 of the main text.

$$L_3 = \sum_{k=1}^{K} \mathbb{E}_{\theta, \Delta\theta} \|h_k(\theta + \Delta\theta) - \exp(C_k\Delta\theta)h_k(\theta)\|^2, \tag{26}$$

$$L_4 = \mathbb{E}\|I - G_\beta(s, v(x), h(\theta))\|^2. \tag{27}$$

$L_3$ is to model the head rotation, and $L_4$ is to model the generation of the posed image.

During training, we alternatively update $(G_\beta, s)$ and $(v(x), B(\theta), u(x'), h(\theta), C)$ by gradient descent on the overall loss function that is a linear combination of $L_0$, $L_1$ and $L_2$ in the main text, as well as $L_3$ and $L_4$ introduced above.

The learned model enables two useful applications:

(a) **Novel view synthesis**. Given an unseen pose $(x, \theta)$, the model can predict the corresponding posed image by $G_\beta(s, v(x), h(\theta))$.

(b) **Inference of pose**, i.e., self-position $x$ and head direction $\theta$, from posed image $I$ alone. Specifically, after training the model, we can learn an additional inference network $F_\xi$ that maps an observed posed image $I$ to its pose embedding $v(x)$ and $h(\theta)$. The inference network is learned by minimizing the $\ell_2$ distance between the predicted and true pose embeddings: $\mathbb{E}\|(v(x), h(\theta)) - F_\xi(I)\|^2$. Then

Table 1: Average error of pose inference.

|  | $x_1$ | $x_2$ | $\theta$ |
| --- | --- | --- | --- |
| Error | .0225m | .0230m | 1.37º |

given an unseen posed image $I$, we can infer the pose by $\arg\min_{x,\theta} \|(v(x), h(\theta)) - F_\xi(I)\|^2$. In this task, $F_\xi(I)$ is the estimate of $(v(x), h(\theta))$, and it is likely that this estimate contains error. This error will translate to the error in the estimated $(x, \theta)$. Thus our theoretical analysis of error translation in the main text is highly relevant, and the isotropic scaling condition is motivated by the analysis of error translation.

We conduct experiments on a dataset generated by the Gibson Environment [14], which provides tools for rendering images of different poses in 3D rooms. Specifically, we select 20 areas of size 2m × 2m from different rooms and render about 28k 64 × 64 RGB posed images for each area. The camera height is fixed and the camera can only rotate horizontally. The scene embedding vector $s$ is of 512 dimensions. Both $v(x)$ and $h(\theta)$ are of 192 dimensions, partitioned into $K = 16$ modules.

Hexagon patterns still emerge in the learned $v(x)$ (average gridness score 0.71). For novel view synthesis, we evaluate the performance on 374k testing posed images. The resulting peak signal-to-noise ratio (PSNR) between synthesized images and ground truth images is 25.17, indicating that the model can generate reasonable unseen posed images. Fig. 12 demonstrates several examples of the novel view synthesis results.

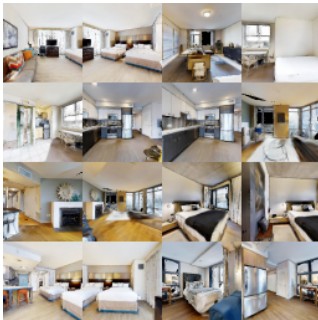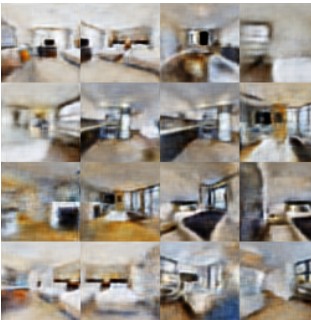

Figure 12: Examples of synthesizing novel views. *Left*: Ground truth unseen posed images. *Right*: synthesized unseen posed images.

For inference of pose (self-position $x = (x_1, x_2)$ and head direction $\theta$), we evaluate the performance on the same 374k testing posed images and report the average inference error in Table 1. The estimates are reasonably accurate.

## 2.7   Ablation studies

**Isotropic scaling condition is necessary for hexagon grid patterns.**   A natural question is whether the isotropic scaling condition (condition 2) is important for learning hexagon grid patterns. To verify this, we learn the model by removing the loss term $L_2$ (Eq. (19) in the main text) from the loss function, which constrains the model to meet condition 2. As shown in Fig. 13, more strip-like patterns emerge without $L_2$, indicating that condition 2 is important for hexagon grid patterns to emerge.

**Assumption of $u(x') \geq 0$ is not necessary for hexagon grid patterns.**   During training, we make an assumption of $u(x') \geq 0$ to make sure the connections from grid cells to place cells are excitatory [16, 10]. However, we want to emphasize this is not a key assumption in our model. Fig. 14 demonstrates the learned neurons in the network without assuming $u(x') \geq 0$, where hexagonal grid firing patterns also emerge. The average gridness score is 0.82 and the percentage of grid cells is 87.50%. However, the grid activations can be either positive/excitatory (in red color) or negative/inhibitory (in blue color).

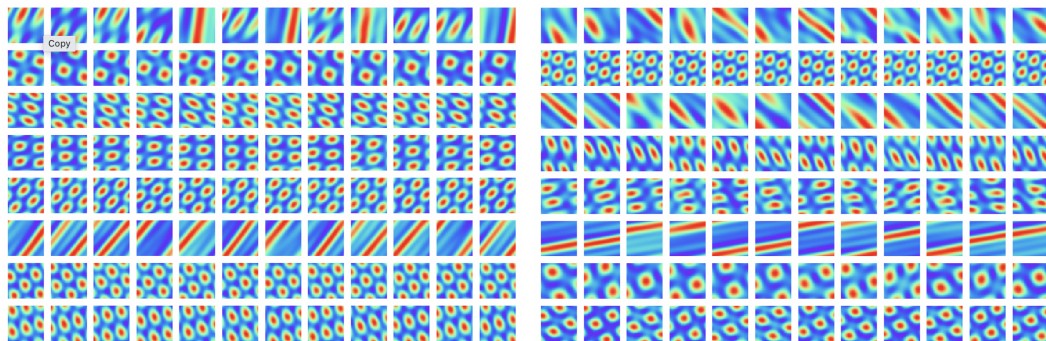

Figure 13: Learned neurons without loss term $L_2$, which is the constraint on isotropic scaling condition. More strip-like firing patterns emerge.

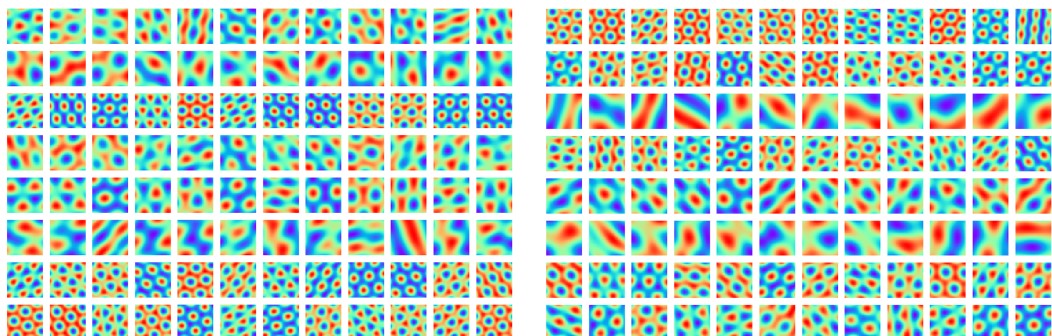

Figure 14: Learned neurons without the assumption of $u(x') \geq 0$. Hexagonal grid firing patterns also emerge, with the grid activations being either positive/excitatory (in red color) or negative/inhibitory (in blue color).

**Skew-symmetric assumption of $B(\theta)$ is not important for hexagon grid patterns.** To make the linear transformation a rotation, we have assumed that $B(\theta)$ is skew-symmetric, i.e., $B(\theta) = -B(\theta)^\top$. Nonetheless, this assumption is not important for the emergence of hexagon grid patterns. Fig. 15 demonstrates the learned neurons without assuming that $B(\theta)$ is skew-symmetric. Hexagon grid firing patterns emerge in most of the neurons, with only one block of square grid firing patterns.

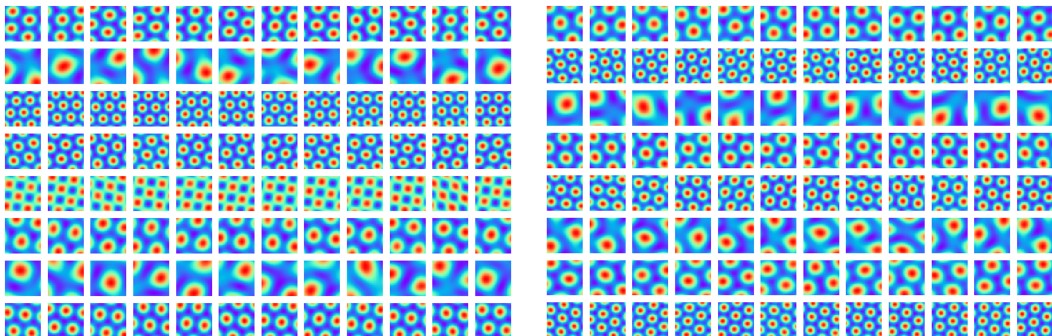

Figure 15: Learned neurons without skew-symmetric assumption of $B(\theta)$. Hexagonal grid firing patterns emerge in most of the neurons, with a block of square grid firing patterns.

**Number and sizes of blocks do not matter.** It is worthwhile to mention that the emergence of hexagonal grid firing patterns in the learned neurons are not due to specific design of the block size or the number of blocks. Fig. 16 visualizes the learned neurons by fixing the total number of neurons at 192 and altering the block size and number of blocks. Hexagon patterns emerge in all the settings.

**Multiple blocks or modules are necessary for learning grid patterns of multiple scales.** We further try to fully remove the assumption of blocks or modules; i.e., we learn a single block of $B(\theta)$. Fig. 17 shows the learned neurons and the corresponding autocorrelograms. All the learned neurons share similar large scales, which indicates that the high frequency part of $A(x, x')$ may not be fitted very well.

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

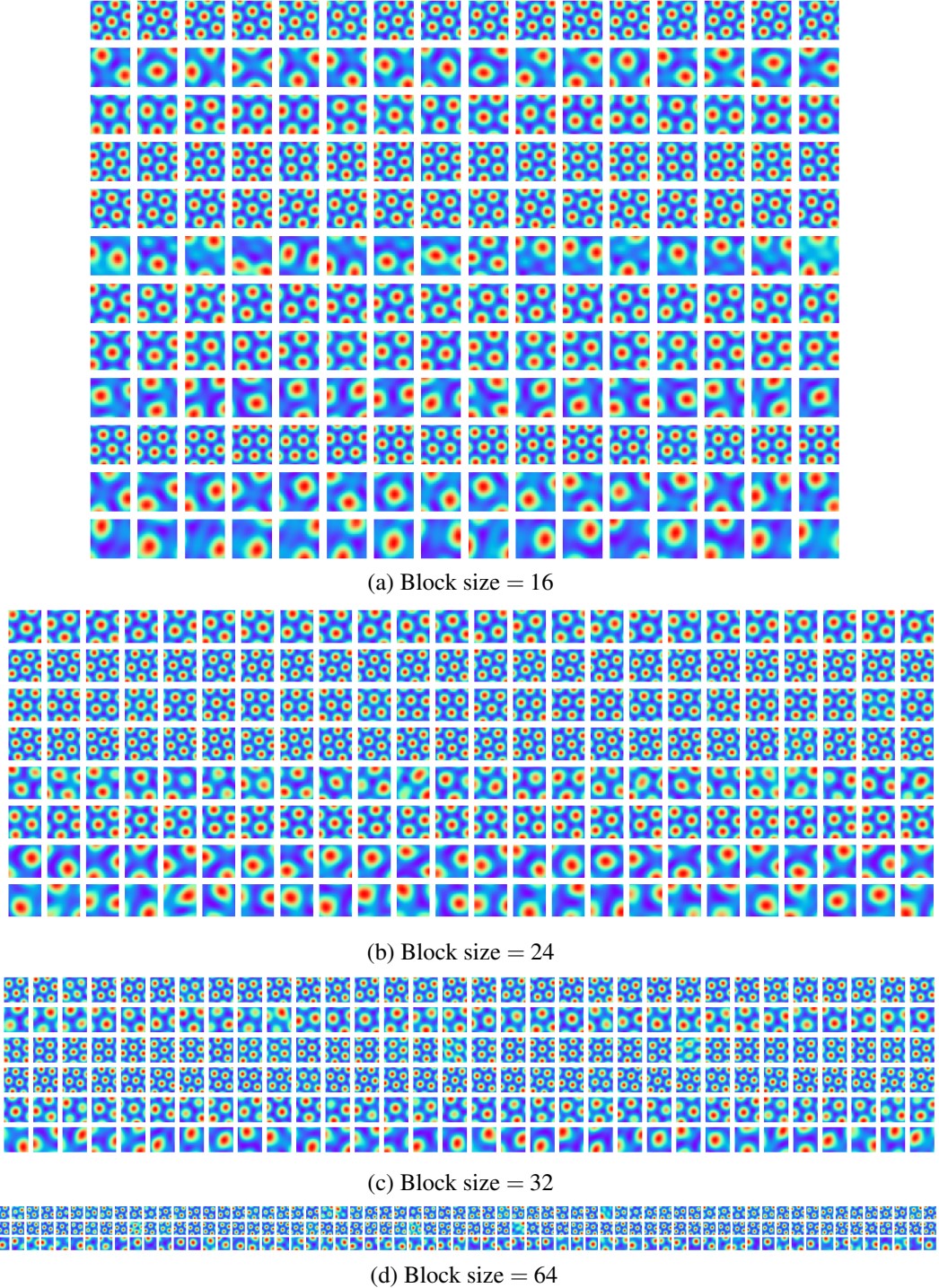

(a) Block size = 16

(b) Block size = 24

(c) Block size = 32

(d) Block size = 64

Figure 16: Learned patterns of $v(x)$ with different block sizes. The total number of units is fixed at 192. Every row shows the learned patterns within the same block.

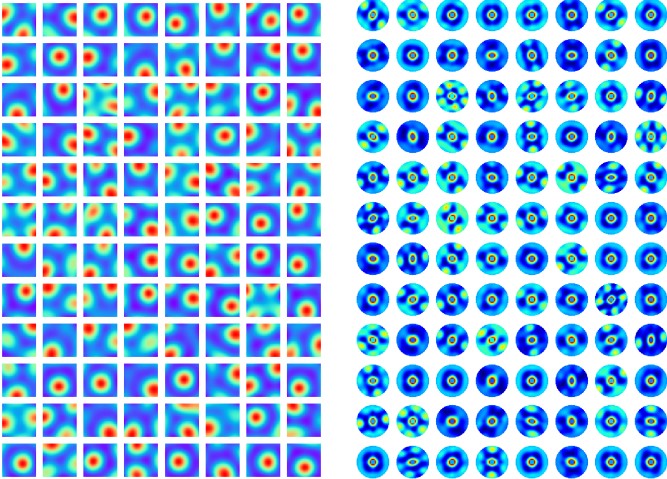

Figure 17: *Left*: learned neurons with a single block of $B(\theta)$. The firing patterns has a single large scale, meaning that the high frequency part of $A(x, x')$ is not fitted very well. *Right*: autocorrelograms of the learned neurons. Some exhibit clear hexagon grid patterns, while the other do not, probably because the scale of those grid patterns are beyond the scope of the whole area.