# OpenReview forum: "On Path Integration of Grid Cells: Group Representation and Isotropic Scaling"
_NeurIPS.cc/2021/Conference — NeurIPS 2021 Poster_

### Official Review · Reviewer_Lump · 2021-07-12

**Rating:** 7
**Confidence:** 3

**Summary:**

The paper introduces a general framework for modeling 2D navigation via grid cells. The main idea is to embed a 2D position x into a higher-dimensional positional encoding v(x) and model 2D translations via group transformations in this embedding space. A specific case are linear transformations via rotation matrices that act on the embedding v(x). Under this assumption, it can be shown that hexagonal patterns which are characteristic for neural grid cells naturally emerge as an optimal encoding.

**Limitations And Societal Impact:**

There is a small discussion on limitations in Sec. 8 but no dedicated section or paragraph.

**Main Review:**

1. Presentation: According to the submission history, the ICML AC had concerns about the exposition and presentation of the paper. The presentation of the paper in its current form is very clear. I find the structure of the methods part adequate (Sec. 2 - Sec. 5) and easy to follow.

2. Quality of theoretical contribution/mathematical validity: Overall, the theory sections are mathematically sound and the stated assumptions are reasonable (l. 82, l. 107), save for a number of minor flaws. It should be straightforward to address these issues in a minor revision:
- The defined set of transformations F are intimately related to the standard mathematic concept of a "flow". Curiously, the authors do not mention this connection. This would also provide a stronger justification for Condition 1 (l. 82).
- Why does the angle Phi not depend on positional differential dx. In Eq. (5), the expansion of dv would envolve both dr and dPhi. If this is an implicit requirement, it should be stated as an explicit assumption.
- What is the motivation for restricting the approach to 2D environments? One could easily conceive of a formulation for more general Euclidean domains, like R^3. Most of the theory should be analogous.

3. Results:
- It is fascinating that the hexagonal patterns emerge from the optimization framework. It's also a strong justification for the validity of the proposed method and assumptions. Furthermore, this effect is validated quantitatively in Table 1.
- It is a little disappointing that all evaluations focus on linear transformation models. One of the motivations of the proposed framework is that it allows for more general group transformations. This should be shown empirically as a proof of concept.
- The path integration experiments lack baseline comparisons which makes it hard to put these results into context. I would like to see comparisons to recurrent [3] and PCA based [33] models in Figure 6.
- The presented ablation study (l. 289) is useful to assess under which circumstances the hexagonal patterns emerge.

Minor remarks:
- I believe that the condition of F(v, 0, Phi) = v (l. 101) is central to the proposed framework and should be stated as a part of "Condition 1" (l. 82).
- The embeddings v(x) lie on the Lie group manifold characterized by rotation matrices. This means that they are contained in an elliptic space with positive curvature. This seems counterintuitive, since the goal is to model 2D trajectory prediction in Euclidean space. E.g. there might be group transformations, such that 3 consecutive steps with 90 degree turns between them yield a closed loop. Is this a phenomenon you encountered or did you find that this effect is relevant in practice?

**Time Spent Reviewing:**

3h.

---

> ### Author Response · Authors · 2021-08-10
> **Response to Reviewer Lump**
>
> Thank you for the detailed review and positive feedback. Also thank you for your positive evaluation of the presentation and mathematical soundness of our paper.
>
> Below we address specific points.
>
> **Q: Transformation F are intimately related to “flow” and such connection should be mentioned.**
>
> A: Thank you for this insightful point! We will include a discussion about flow in the revision.
>
> **Q: Why does the angle not depend on positional differential $\delta x$?**
>
> A: That’s a good question. When doing the Taylor expansion, we treat $v(x + \delta x) = F(v(x), \delta r, \theta)$ as a function of a single variable $\delta r$ (which is infinitesimal), while treating $\theta$ as fixed. We shall make this point more explicit in revision as you suggested.
>
> **Q: One could easily conceive of a formulation for more general Euclidean domains, like R^3.**
>
> A: That’s a great suggestion! Indeed we are planning to apply our model to the 3D environment to study 3D grid cells such as those in bats.
>
> **Q: One of the motivations of the proposed framework is that it allows for more general group transformations. This should be shown empirically as a proof of concept.**
>
> A: Following your advice, we tested our model with a nonlinear transformation model:
>
> $$F(v(x), \Delta r, \theta) = {\rm ReLU}(\exp(B(\theta) \Delta r) v(x)).$$
>
> Interestingly, regular hexagon patterns also emerge (average gridness score 0.83, percentage of grid cells 70.21%). We will include the results in the revised manuscript.
>
> **Q: I would like to see comparisons of path integration experiments to recurrent [3] and PCA based [33] models in Figure 6.**
>
> A: Following your advice, we compare with models of [3] (LSTM) and [33] (RNN) using the code released by [33]. To make the results fully comparable, we re-train and test their models using trajectories simulated in our environment, which is slightly different from their original environments in terms of area size and distribution of velocity.  We summarize the path integration performances at different time steps in the following table. All results are computed over the same 1,000 testing trajectories.
>
> | Time step          | 100                      | 200                     | 300                     | 400                     | 500                     | Avg                     |
> |--------------------|--------------------------|-------------------------|-------------------------|-------------------------|-------------------------|-------------------------|
> | LSTM               | 0.062 $\pm$ .095         | 0.138 $\pm$ .155        | 0.176 $\pm$ .157        | 0.189 $\pm$ .167        | 0.241 $\pm$ .177        | 0.138 $\pm$ .151        |
> | RNN                | 0.045 $\pm$ .025         | 0.085 $\pm$ .047        | 0.123 $\pm$ .063        | 0.165 $\pm$ .079        | 0.200 $\pm$ .103        | 0.105 $\pm$ .082        |
> | Ours w/o re-encode | 0.027 $\pm$ .018         | 0.037 $\pm$ .022        | 0.046 $\pm$ .025        | 0.051 $\pm$ .027        | 0.055 $\pm$ .028        | 0.038 $\pm$ .027        |
> | Ours w/ re-encode  |  $7.59e^{-5}$ $\pm$ .001 | $7.59e^{-5}$ $\pm$ .001 | $1.01e^{-4}$ $\pm$ .002 | $1.27e^{-4}$ $\pm$ .002 | $1.77e^{-4}$ $\pm$ .002 | $9.69e^{-5}$ $\pm$ .002 |
>
> We will include the above table in the revised manuscript.
>
> **Q: I believe that the condition of F(v, 0, Phi) = v (l. 101) is central to the proposed framework and should be stated as a part of "Condition 1" (l. 82).**
>
> A: Thanks for the insightful and important point! You are absolutely right. We will revise Condition 1 to include that equation as you advised.
>
> **Q: The embeddings $v(x)$ lie on the Lie group manifold characterized by rotation matrices … did you find that this effect is relevant in practice?**
>
> A: Thanks for the interesting and insightful question. About the phenomenon you mentioned, we have not encountered it in our experiments. One possible explanation is as follows:  let $v(x)$ be d-dimensional representation with constant $||v(x)||$. Then $(v(x), \forall x)$ forms a 2D manifold on the $(d-1)$ sphere. However, when the agent moves on a straight line segment in the 2D Euclidean space, $v(x)$ does not necessarily move on a great circle or geodesic on the $(d-1)$ sphere. Conversely, a geodesic path on the $(d-1)$ sphere does not necessarily belong to the 2D manifold $(v(x), \forall x)$. Thus the edges of a triangle on the $(d-1)$ sphere with 90 degree angles do not necessarily belong to the 2D manifold either.
>
> **Summary**
>
> We will improve and revise our manuscript and include the new results we obtained by following your insightful suggestions.

---

### Official Review · Reviewer_G4HX · 2021-07-16

**Rating:** 5
**Confidence:** 5

**Summary:**

This paper focuses on the find that how the grid cells perform path integration. Speciafically, this paper introduces the isotropic scaling condition to produce the hexagon firing pattern for grid cells. The optimization experiments show that the proposed formulation of place and grid cells demonstrated this.

**Limitations And Societal Impact:**

It seems that the only contribution is to introduce the isotropic scaling to generate the hexagon pattern during path integration. However, why should we do this? What ‘s the necessity to connect hexagon pattern and path integration via optimization, if predefined cosine functions can generate hexagon pattern very well?

**Main Review:**

Although I appreciated that the authors provide an interesting formulation for the relationship of place and grid cells, I have the following concerns:

1.	The authors claim that they don’t make any orthogonality assumption and the isotropic scaling guarantees the l_2 error in L122. However, in the proof of Theorem 3 of this error bound, explicit orthogonal assumptions are made. I expect the authors to provide, in the rebuttal, a more rigorous proof for this error bound without any orthogonal assumptions.

2.	Many literatures have shown that cos/sin functions can produce hexagon patterns, e.g., [a, b]. Can you provide more connections to those results, because I highly suspect that the three conditions introduced in this paper will result in the cosine functions (Maybe the formulation (e.g., the isotropic scaling) in this paper is mathematically equivalent to the cosine functions?), especially after seeing the Fig 7 in Appendix. If this conjecture is correct, then I see no contribution for this paper, since we can simply use the predefined functions as in [a] to do both path integration and generate hexagon. Then what’s the purpose for the optimization?
3.	The authors didn’t show that the linear model with isotropic condition has a hexagon grid pattern in the proof, thus the tittle of Sec 3.3 is misleading, “hexagon grid pattern” in the section title should be removed.
4.	Theoretic guarantee for generating the hexagon pattern by the introduced formulation will strengthen this paper a lot.
5.	It’s not clear how to derive the third item from the second item in Eq (9).

[a] Burgess N, Barry C, O'keefe J. An oscillatory interference model of grid cell firing. Hippocampus. 2007 Sep;17(9):801-12.

[b] Fuhs MC, Touretzky DS. A spin glass model of path integration in rat medial entorhinal cortex. Journal of Neuroscience. 2006 Apr 19;26(16):4266-76.



**Time Spent Reviewing:**

6

---

> ### Author Response · Authors · 2021-08-10
> **Response to Reviewer G4HX**
>
> Thank you for your review and valuable comments. Below we address specific questions.
>
> **Q: About orthogonal assumption in the proof of Theorem 3.**
>
> A: We wish to clarify a factual misunderstanding about orthogonality. The orthogonality between the basis functions $(v_i(x), i=1,..., d)$ in Section 4 and the orthogonality in the proof of Theorem 3 are two different things.
>
> For the proof of Theorem 3, in Theorem 1, we have proved that the 2D local neighborhood around a self-position $x$ in the 2D physical space is embedded conformally as a local 2D plane around $v(x)$ in the $d$-dimensional neural space. For any 2D plane in the d-dimensional space, it is an **intrinsic property** that there exists an orthogonal basis $(v_1, v_2)$ to span the plane, which is not an explicit assumption that we make. In the proof of theorem 3, we use this property to show that $\mathbb{E}||\delta v||^2 = 2\tau^2$, and the orthogonal basis $(v_1, v_2)$, which always exists, has nothing to do with the representation we propose to learn.
>
> In contrast, the PCA-based basis expansion model (Section 4) assumes that the basis functions $v(x) = (v_i(x), i=1,…,d)^\top$ (which are $d$ functions of $x$) are orthogonal basis functions for expanding $A(x, x’)$ (which is function of $x$ for each $x’$). This is a strong explicit assumption added on the learned representation $v(x)$. We do not make such an explicit orthogonal assumption.
>
> **Q: Figure 7 in appendix.**
>
> A: We wish to make a clarification. Figure 7 in supplementary is about sine/cosine tunings of $B(\theta)$ over direction $\theta$, which confirms our Theorem 1. It is not about sine/cosine tunings of $v(x)$, which consists of 2D response maps shown in Figure 3.
>
>  **Q: Connection to References [a, b]**
>
> A: Thank you for bringing up these references. Ref [a] (Burgess, Barry, O'keefe, Hippocampus, 2007) describes an oscillatory interference model of the grid cell formation.  Part of Ref [b] (Fuhs & Touretzky, 2006) presents a descriptive model of grid cell responses (their Eq. 11) by summing up three cosine functions with an orientation offset of 60 deg, which is similar to Solstad et al (Hippocampus, 2006) for modeling the transformation from grid cells to place cells. Refs [a,b]  provide valuable initial modeling insights on how grid response patterns could be described quantitatively, and how the grid response patterns might be generated mechanistically via either interference of oscillatory patterns or attractor-based mechanisms.
>
> As we will elaborate in more detail in response to the next question, our model represents a different type of models compared to models in Refs [a,b]. Our model can be thought of as a normative model, which starts from first principles to provide the rationale behind the grid responses, as Reviewer Wn1K pointed out.  We consider these different types of models all have their values in understanding the grid cells and the brain's spatial navigation system. We will add a discussion on this point in the revised manuscript.
>
> **Q: The purpose of optimization if predefined cosine functions can generate hexagon pattern.**
>
> A: While descriptive models (like References [a, b]) use predefined cosine functions to characterize hexagon patterns of grid cells and path integration, they do not explain where the cosine functions come from. In contrast, our method can be considered a normative model where we seek a deeper understanding and explanation of the emergence of hexagon patterns in grid cells. To this end, we only recruit generic vectors and matrices in our model and identify two minimally simple conditions on the transformation of path integration. Results obtained by our numerical optimization agree with various neuroscience observations.
>
> Our optimization-based approach follows the recent line of research pioneered by [3, 4] which studies grid cells by machine learning models. Using generic vectors and matrices, our method has the advantage that it can be potentially applied to more general situations where it is difficult to predefine functions for neuronal activities. For example, [1, 2] indicate that the patterns of grid cells would deform due to irregular geometry or the presence of rewards or obstacles. In such cases, we may first learn the deformed kernel functions of place cells ($A(x, x’)$) using empirical trajectories of the agent as in [2], which then leads to the update of grid cells $v(x)$ in our model, enabling us to study the deformed grid cells. Another general situation is to learn grid cells in three-dimensional space, as suggested by Reviewer Lump, which may have implications for studying the navigation system in animals such as bats. Our modeling scheme may also be generalized to represent poses and their continuous transformations in general, e.g., the poses of arms of the agent, the poses of faces (or other objects) in a 3D visual scene. We plan to investigate these problems in future work.
>
> The reviewer also mentioned that "I highly suspect that the three conditions introduced in this paper will result in the cosine functions". This is an interesting question. It is mathematically highly non-trivial to obtain the general closed-form solutions to our model (i.e., equations 15, 16, 17), and the solutions can be more complex than the superposition of three cosine waves in References [a, b]. Following the suggestion of Reviewer Lump, we have also experimented with a nonlinear transformation model, where a ReLU activation is included:
>
> $$F(v(x), \Delta r, \theta) = {\rm ReLU}(\exp(B(\theta) \Delta r) v(x)).$$
>
> We are still able to learn hexagon patterns using numerical optimization (average gridness score 0.83, percentage of grid cells 70.21%). The learned response maps have regions with effectively zero responses, suggesting that the solutions are not simple cosine waves. In general, our framework can lead to richer consequences and can account for more general situations than cosine functions.
>
> Finally, if one could indeed show a mathematical connection between our assumptions and the cosine functions under certain conditions, that would be an interesting result as well, because it provides a reason why cosine functions should be favored.
>
> **Q: About linear model / introduced formulation and hexagon grid patterns.**
>
> A: We can show that, if the response maps of each module have hexagon patterns and are shifted versions of each other, then they can satisfy the basis expansion model, the transformation model, and the isotropic scaling condition approximately. We thus far rely on numerical optimization to show that the converse is true, i.e., the optimization of our loss function (equations 15, 16, 17) leads to such response maps. Relying on numerical optimization is a practice shared by other optimization-based methods [3, 4, 5, 6]. We will continue to push the theoretical analysis and we will revise the title of Section 3.3 by following your advice.
>
> **Q: how to derive the third term from the second term in Eq (9)**
>
> A: It is a property or one definition of the matrix exponential, which generalizes the exponential of real numbers. See Property 4 in [7].
>
> **Q: why isotropic scaling.**
>
> A: We justify isotropic scaling by error analysis in Theorem 3 in the main text and Proposition 1 in supplementary. Theorem 1 shows that isotropic scaling leads to locally conformal embedding. These results provide strong rationales for isotropic scaling.
>
> **Summary**
>
> With the above technical discussions, we wish to be allowed to engage in a more general discussion about descriptive vs normative models because this seems your main concern with our contribution. Great examples of descriptive models include Kepler's ellipses for planets and de Broglie's sinusoidal or periodic waves for electrons. No doubt they are revealing discoveries and profound insights, but it is still worthwhile to pursue normative models such as Newtonian mechanics and Schrodinger equation. By no means we are comparing our humble attempt to these monumental masterpieces, all we are trying to say is that studying normative models is worthwhile as they may provide deeper explanations and may be more generally applicable.
>
> We will revise and improve our paper to address your criticisms. We hope our paper will be interesting to the NeurIPS community. Thank you for your valuable comments and we respectfully request you to reconsider our paper in view of our reply.
>
>
>
> [1] Butler, William N., Kiah Hardcastle, and Lisa M. Giocomo. "Remembered reward locations restructure entorhinal spatial maps." Science (2019).
>
> [2] Stachenfeld, Kimberly L., Matthew M. Botvinick, and Samuel J. Gershman. "The hippocampus as a predictive map." Nature neuroscience (2017).
>
> [3] Banino, Andrea, et al. "Vector-based navigation using grid-like representations in artificial agents." Nature (2018).
>
> [4] Cueva, Christopher J., and Xue-Xin Wei. "Emergence of grid-like representations by training recurrent neural networks to perform spatial localization." ICLR (2018).
>
> [5] Sorscher, Ben, et al. "A unified theory for the origin of grid cells through the lens of pattern formation." NeurIPS (2019).
>
> [6] Cueva, Christopher J., et al. "Emergence of functional and structural properties of the head direction system by optimization of recurrent neural networks." ICLR (2019).
>
> [7] Salman, Mohammed Abdullah Saleh, and Dr VC Borkar. "Exponential Matrix and Their Properties." International Journal of Scientific and Innovative Mathematical Research (IJSIMR) (2016).

---

> > ### Comment · Reviewer_G4HX · 2021-08-26
> > **Reply**
> >
> > Thanks for your reply.
> >
> > My concerns on (1) and (5) are resolved. The authors admitted (4).  However, my concerns on (2)(4) still exist.
> >
> > I was not saying Figure 7 is v(x). I am not denying normative models. I just think that the usefulness of this optimization is not convincing. This is different from [3] in which the objective is navigation, and the hexagon grid patterns are just by-products. However, it seems that the only objective in this paper is to generate the hexagon grid patterns. If the authors believe that it can be applied to other useful applications, experimental results are needed.
> >
> > I will not feel uncomfortable if this paper is accepted, however, I **strongly disagree** with that this paper can be accepted as "10: Top 5% of accepted NeurIPS papers, seminal paper". It can be weakly accepted.

---

> > > ### Author Response · Authors · 2021-08-26
> > > **Thank you for your comments**
> > >
> > > Thank you for your further comments, and for your judgement that our paper can be weakly accepted. We will continue to improve our paper based on your criticisms.
> > >
> > > The main objective of our work is to understand how path integration can be accomplished by the grid cells system, and our experiments show that the learned model is capable of accurate path integration even in the presence of noises. We will try our best to add experiments to further illustrate useful applications of our method.
> > >
> > > Again, we thank you for your valuable comments.

---

> > > ### Author Response · Authors · 2021-09-01
> > > **Reporting additional experiments on egocentric vision and path planning**
> > >
> > > **Following your suggestion on adding experiments on applications**, we conduct the following two experiments and have obtained initial results. The first experiment is to integrate our grid cells model with egocentric vision. The second experiment is on path planning. The two experiments further illustrate the usefulness of our model, in addition to the path integration experiments in our paper. The following are details of the additional experiments. Thank you for your time and consideration.
> > >
> > > **(1) Integrating egocentric vision.**
> > >
> > > When the agent moves in darkness, it can infer its self-position by integrating self-motion, as illustrated by our experiments on path integration. If there is visual input, the agent can infer its self-position (as well as head direction) from the visual image alone. We extend our grid cells model to study this problem of egocentric vision, which is important in computer vision.
> > >
> > > Specifically, suppose the agent navigates in a 3D scene such as a room, and the height of the eye (or camera) remains fixed. Suppose at 2D self-position $x$ and with head direction $\theta$, the agent sees an image ${\rm I}$, which is called a posed image. We use the vector representation $v(x)$ in our original grid cells model to represent the 2D self-position $x$, and use another vector representation $h(\theta)$ to represent the head direction $\theta$. If the agent changes its head direction from $\theta$ to $\theta + \Delta \theta$, $h(\theta)$ is transformed to
> > >
> > > $h(\theta + \Delta \theta) = \exp(C\Delta\theta)  h(\theta)$.
> > >
> > > We assume that there are $K$ modules or blocks in $h(\theta)$ and $C$ is skew-symmetric. This is similar to the transformation of $v(x)$ in our grid cells model.
> > >
> > > $(x, \theta)$ is called the pose of the camera (or eye), and we call $(v(x), h(\theta))$ the pose representation.
> > >
> > > To associate the pose representation $(v(x), h(\theta))$ with the posed image ${\rm I}$, we use a vector representation $c$ to represent the 3D scene which is shared across different posed images of the same scene, and we learn a generator network $G_\beta$ that maps $c$ and $(v(x), h(\theta))$ to the posed image ${\rm I}$:
> > >
> > > ${\rm I} = G_\beta(c, v(x), h(\theta))$,
> > >
> > > where the generator $G_\beta$ is parametrized by a multi-layer deconvolutional neural network with parameters $\beta$.
> > >
> > > Given the above assumptions, we introduce two extra loss terms in addition to the loss function described in Section 5 (Eqn 16-18):
> > >
> > > $L_3 = \sum_{k=1}^K \mathbb{E}_{\theta, \Delta \theta} ||h_k(\theta + \Delta \theta) - \exp(C_k \Delta \theta)h_k(\theta)||^2$.
> > >
> > > $L_4 = \mathbb{E} ||{\rm I} – G_\beta(c, v(x), h(\theta))||^2$.
> > >
> > > $L_3$ is to model the head rotation, and $L_4$ is to model the generation of the posed image.
> > >
> > > During training, we alternatively update $(G_\beta, c)$ and $(v(x), B(\theta), u(x’), h(\theta), C)$ by gradient descent on the overall loss function that is a linear combination of $L_0$, $L_1$ and $L_2$ in our paper, as well as $L_3$ and $L_4$ introduced above.
> > >
> > > The learned model enables two useful applications:
> > >
> > > **(a) Novel view synthesis**. Given an unseen pose $(x, \theta)$, the model can predict the corresponding posed image by $G_\beta(c, v(x), h(\theta))$.
> > >
> > > **(b) Inference of pose**, i.e., self-position $x$ and head direction $\theta$, from posed image ${\rm I}$ alone. Specifically, after training the model, we can learn an additional inference network $F_\xi$ that maps an observed posed image ${\rm I}$ to its pose representation $v(x)$ and $h(\theta)$. The inference network is learned by minimizing the $\ell_2$ distance between the predicted and true pose representations: $\mathbb{E}||(v(x), h(\theta)) – F_\xi({\rm I})||^2$. Then given an unseen posed image ${\rm I}$, we can infer the pose by $\arg \min_{x, \theta}||(v(x), h(\theta)) – F_\xi({\rm I})||^2$. In this task, $F_\xi(I)$ is the estimate of $(v(x), h(\theta))$, and it's likely that this estimate contains error. This error will translate to the error in the estimated $(x, \theta)$. Thus our theoretical analysis of error translation in the paper is highly relevant, and the isotropic scaling condition is motivated by the analysis of error translation.
> > >
> > > We conduct experiments on a dataset generated by the Gibson Environment [1], which provides tools for rendering images of different poses in 3D rooms. Specifically, we select 20 areas of size 2mx2m from different rooms and render about 28k 64x64 RGB posed images for each area. The camera height is fixed and the camera can only rotate horizontally. The scene representation vector $c$ is of 512 dimensions. Both $v(x)$ and $h(\theta)$ are of $192$ dimensions, partitioned into $K=16$ modules.
> > >
> > > Hexagon patterns still emerge in the learned $v(x)$ (avg. gridness score 0.71). For novel view synthesis, we evaluate the performance on 374k testing posed images. The resulting peak signal-to-noise ratio (PSNR) between synthesized images and ground truth images is 23.53, indicating that the model can generate reasonable unseen posed images.
> > >
> > > For inference of pose (self-position $x = (x_1, x_2)$ and head direction $\theta$), we evaluate the performance on the same 374k testing posed images. The results are summarized below.
> > >
> > > |       | $x_1$   | $x_2$   | $\theta$       |
> > > |-------|---------|---------|----------------|
> > > | Error | 0.0225m | 0.0230m | 1.37$^{\circ}$ |
> > >
> > > We will include the results, examples of synthesized posed images and patterns of learned $v(x)$ in the revision.
> > >
> > >
> > > **(2) Path planning.**
> > >
> > > Our grid cells model can be applied to path planning. Specifically, according to [2], the adjacency kernel can be modeled as
> > >
> > > $ A_\gamma(x, x') = \mathbb{E} \left[ \sum_{t=0}^{\infty} \gamma^t 1(x_t = x') | x_0 = x)\right] = \langle v(x), u_\gamma(x')\rangle$,
> > >
> > > where $\gamma$ is the discount factor that controls the temporal and spatial scales, $\mathbb{E}$ is with respect to a random walk exploration policy and $1(\cdot)$ is the indicator function. For random walk in open field, $A_\gamma(x, x') \propto \exp(-|x-x'|^2/2\sigma_{\gamma}^2)$, where $\sigma_{\gamma}^2$ depends on $\gamma$.
> > >
> > > To enable path planning, we need kernels of both big and small spatial scales to account for long and short distance planning respectively. To this end, we discretize $\gamma$ into a finite list of scales, and learn a list of corresponding $u_\gamma(x’)$ together with $v(x)$ and $B(\theta)$ using the loss function in Section 5 (Eqn. 16-18).
> > >
> > > With the learned model, we propose the following method for path planning. Let $\hat{x}$ be the target or destination. Let $x^{t}$ be the current position in the path planning process, encoded by $v(x^{t})$, the agent plans the next displacement by
> > >
> > > $\Delta x^{t+1} = \arg \max_{\Delta x } \langle M(\Delta x) v(x^{t}), u_\gamma(\hat{x})\rangle$,  --- (*)
> > >
> > > and then $x^{t+1} = x^{t} + \Delta x^{t+1}$. $M(\Delta x) = \exp(B(\theta) \Delta r)$, where $\Delta x = (\Delta r \cos \theta, \Delta r \sin \theta)$. The idea is that we plan the next step to maximally increase the adjacency to the target: $A_\gamma(x^{t}+\Delta x, \hat{x}) = \langle v(x^{t}+\Delta x), u_\gamma(\hat{x})\rangle = \langle M(\Delta x) v(x^{t}), u_\gamma(\hat{x})\rangle$. $\Delta x$ is chosen from all the allowed displacements for a single step, and $\gamma$ is selected as the smallest one that satisfies $\max_{\Delta x} \langle M(\Delta x)v(x^t), u_\gamma(\hat{x}) \rangle > 0.2$.
> > >
> > > We test path planning in the open field environment. The model is first learned using a single-scale kernel function $A_\gamma(x, x’) = \exp(-|x-x'|^2/2\sigma_{\gamma}^2)$ where $\sigma_\gamma = 0.07$. Then we assume a list of three scales: $\sigma_\gamma = [0.07, 0.14, 0.28]$ and learn the corresponding list of $u_\gamma(x’)$. The pool of allowed displacements for a single step is defined as:  $dr$ can be 1 or 2 grids, while $\theta$ can be chosen from 200 discretized angles over $[0, 2\pi]$. Our experimental results show that the agent is able to plan straight path to the target in the open field environment. When $x^{t}$ is far from the target, kernel with large $\sigma_\gamma$ is chosen, and as $x^{t}$ approaches the target, the chosen kernel gradually switches to the one with small $\sigma_\gamma$. A planning episode is treated as a success if the distance between $x^{t}$ and target is smaller than $0.5$ grid within $40$ time steps. The agent achieves a success rate of 100% (tested for $10,000$ episodes). We will include the results and examples of planned paths in the revision.
> > >
> > > For a field with obstacles or rewards, we can learn the deformed $A_\gamma(x, x’)$ and $(v(x), u_\gamma(x’))$ by temporal difference learning with a random walk exploration policy as suggested in [2]. After learning $A_\gamma(x, x’)$ and $(v(x), u_\gamma(x’))$, we can continue to use equation (*) for path planning. We shall further study it in future work.
> > >
> > > **Summary**
> > >
> > > We believe that the above two experiments are substantial additions to the existing experiments in our paper, and they are highly relevant to navigation.
> > >
> > > Thank you again for your consideration.
> > >
> > >
> > >
> > > [1] Fei Xia, Amir R Zamir, Zhiyang He, Alexander Sax, Jitendra Malik, and Silvio Savarese. Gibson env: Real-world perception for embodied agents. In Proceedings of the IEEE Conference on Computer Vision and Pattern Recognition, pages 9068–9079, 2018.
> > >
> > > [2] Kimberly L Stachenfeld, Matthew M Botvinick, and Samuel J Gershman. The hippocampus as a predictive map. Nature neuroscience, 20(11):1643, 2017.

---

### Official Review · Reviewer_Wn1K · 2021-07-19

**Rating:** 10
**Confidence:** 4

**Summary:**

The paper derives a model of grid cells from first principles, starting with the basic assumption that position is represented as a high-D vector and that it is updated as a function of direction and speed.  The authors stipulate an isotropic scaling condition in order to make the representation robust to neural noise.  The elements of the resulting high-D embedding resemble grid cells, and importantly, the ability for error correction during path integration.


**Limitations And Societal Impact:**

yes

**Main Review:**

I find this to be a super interesting and exciting paper.  There are by now many papers that claim to show how grid cells emerge as a result of some kind of learning rule, but in my view this is the most insightful analysis I've seen to date.  It is a sensible and mathematically grounded approach.  Although they show how the grid cells emerge from learning, the resulting solution corroborates the theory - and so we understand why this is happening.  An important part of the story is robustness to noise, which is clearly linked to the isotropic condition and hence hexagonal grids.  This is high impact work that I believe will of interest to both the NeurIPS and computational neuroscience communities.




**Time Spent Reviewing:**

1.5 hour

---

> ### Author Response · Authors · 2021-08-10
> **Response to Reviewer Wn1K**
>
> Thank you for your very positive evaluation. We are profoundly grateful that you share our excitement about this work.
>
> **About first principles.**
>
> You are absolutely right that we seek to understand grid cells from first principles. Thank you for your precise and deeply insightful summary of our work and contribution. Also thank you for pointing out that our work is mathematically grounded, and that our paper is of interest to the NeurIPS and computational neuroscience communities.
>
> **Robustness to noise.**
>
> Yes, robustness to noise justifies isotropic scaling, which leads to conformal embedding, and which appears to underly hexagon grid patterns.
>
> Thank you again for your deep understanding of our work!

---

### Decision · Program_Chairs · 2021-09-27

**Decision:**

Accept (Poster)

**Comment:**

The reviewers are overall positive. The presented approach is mathematically sound and shows that grid cells can emerge from an optimization framework with minimal assumptions. The main criticism raised is whether the approach is useful beyond already existing models. The authors should clarify potential applications more explicitly in the revision, as well as include the additional experiments they cited in the rebuttal.